# Combining Denoised Neural Network and Genetic Symbolic Regression for Memory Behavior Modeling via Dynamic Asynchronous Optimization

## Abstract

Memory behavior modeling is a critical topic in cognitive psychology and education. Traditional psychological approaches describe the dynamic properties of memory through memory equations derived from experimental data, but these models often lack accuracy and are frequently debated in terms of their form. In recent years, data-driven modeling methods have improved predictive accuracy but often suffer from poor interpretability, limiting their ability to provide deeper cognitive insights. While knowledge-informed neural network models have achieved significant success in fields such as physics, their application in behavior modeling remains limited. This paper proposes a Self-evolving Psychology-informed Neural Network (SPsyINN), which leverages classical memory equations as knowledge modules to constrain neural network training. To address challenges such as the difficulty in quantifying descriptors and the limited interpretability of classical memory equations, a genetic symbolic regression algorithm is introduced to conduct evolutionary searches for more optimal expressions based on classical memory equations, enabling the mutual progress of the knowledge module and the neural network module. Specifically, the proposed approach combines genetic symbolic regression and neural networks in a parallel training framework, with a dynamic joint optimization loss function ensuring effective knowledge alignment between the two modules. Then, for addressing the training efficiency differences arising from the distinct optimization methods and computational hardware requirements of genetic algorithms and neural networks, an asynchronous interaction mechanism mediated by proxy data is developed to facilitate effective communication between modules and improve optimization efficiency. Finally, a denoising module is integrated into the neural network to enhance robustness against data noise and improve generalization performance. Experimental results on four large-scale real-world memory behavior demonstrate that SPsyINN outperforms state-of-the-art methods in predictive accuracy. Ablation studies further show that the proposed approach effectively achieves mutual progress between different modules, improving model predictive accuracy while uncovering more interpretable memory equations, highlighting the potential application value of SPsyINN in psychological research. Our code is released at: https://anonymous.4open.science/r/SPsyINN-3F18

## 1 Introduction

Memory is a crucial component of human cognition and a major focus of research in psychology and neuroscience. Memory behavior modeling aims to establish a relationship model between historical memory behavior and memory performance (e.g., the recall probability for specific materials) to elucidate key patterns of human memory behavior, predict performance, and simulate the forgetting process. These models help researchers better understand the mechanisms of memory and develop effective memory strategies, offering significant academic and practical value (Clark, 2018).

The earliest memory behavior model dates back to 1885 when Ebbinghaus proposed the forgetting curve (Ebbinghaus et al., 1913), suggesting that the relationship between memory performance and time interval follows an exponential function. Subsequently, models such as the generalized power law (Wickelgren, 1974), the adaptive control of thought-rational model (Anderson et al., 2004), and the multi-scale contextual model (Pashler et al., 2009) were introduced. These classical models describe the relationships between memory performance and key memory behavior features (e.g., interval time, repetition frequency, and material difficulty) using mathematical formulas. Derived by experts based on experimental data, these theories lack consensus due to the complexity of memory behavior. Current models often face limitations such as insufficient interpretability, inadequate predictive accuracy, and difficulty quantifying descriptors (Brown & Brown, 2018).

Recently, data-driven approaches have emerged for memory behavior modeling. Techniques like machine learning and deep learning have been extensively applied to large-scale memory behavior datasets, resulting in various parametric models (Settles & Meeder, 2016; Ma et al., 2023; Tu et al., 2020; Liu et al., 2023). These models exhibit significant advantages in predictive accuracy compared to classical theories. However, their complexity makes them difficult to interpret, offering limited theoretical insights. Moreover, data-driven models demand high-quality and large-scale datasets (Rudin, 2022; Wang et al., 2024b), posing additional challenges (Li et al., 2022).

Knowledge-informed neural network models incorporate domain knowledge into neural network construction, enhancing stability and interpretability. These models have achieved remarkable success in natural science tasks (Wang et al., 2024a). For instance, physics-informed neural networks (Raissi et al., 2019) use known equations and boundary conditions as constraints, reducing data dependency and improving both stability and interpretability. However, their application in memory behavior modeling remains limited, primarily due to the insufficient explanatory power of memory knowledge and difficulties in quantifying descriptors. Existing memory equations are neither as precise nor as universally accepted as physical equations for describing or predicting real-world phenomena. Furthermore, abstract descriptors used in classical memory equations, such as memory strength (Wickelgren, 1974) and word difficulty (Lindsey et al., 2014), lack precise formulations, making them challenging to convert into computable variables and complicating knowledge representation.

Based on this analysis, we aim to develop a knowledge-informed neural network model for memory behavior modeling by constraining neural network training using existing memory theory equations to achieve knowledge injection and alignment. To address the limited explanatory power and quantification challenges of classical memory equations, genetic symbolic regression algorithm is introduced. It is initialized with classical memory equations as the population and evolves through mutations to search for improved descriptors and memory equations. Compared to other symbolic regression methods, genetic symbolic regression allows the use of initial equations to fully leverage existing theories and can control equation complexity by limiting symbolic tree depth, ensuring model interpretability. Ultimately, we aim to enable mutual learning and co-optimization between memory equation models and neural networks, enhancing both performance and interpretability.

Based on this framework, we developed a **S**elf-evolving **Psy**chology-**I**nformed **N**eural **N**etwork (SP-syINN), comprising a genetic symbolic regression (GSR) module and a neural network module, with knowledge alignment achieved through interaction and constraint mechanisms. Specifically, we propose a Dynamic Asynchronous Optimization (DAO) method to address dynamic differences during training, including model capability differences and optimization efficiency differences. Model capability differences arise as the GSR module, initialized with classical theories, significantly outperforms the randomly initialized neural network in fitting ability at the start, requiring the neural network to learn more from the GSR module while minimizing its influence on the memory equations. As training progresses, this gap dynamically shifts, so we adjust of different training objectives using a dynamic knowledge alignment method to ensure stable optimization. In addition, optimization efficiency differences stem from genetic symbolic regression relying on CPU-based genetic algorithms, while neural networks leverage gradient-based GPU optimization, which is significantly faster. To address this, we introduce a proxy dataset to facilitate asynchronous knowledge transfer, ensuring flexible interactions, and design multiple asynchronous interaction strategies to enable decoupled module training while achieving efficient knowledge alignment, allowing synchronized co-optimization across both modules.

The main contributions of this paper can be summarized as follows:

- To the best of our knowledge, this is the first work to integrate psychological theories into neural networks for memory behavior modeling. We propose a self-evolving psychology-informed neural network (SPsyINN), consisting of a genetic symbolic regression module and a denoising neural network module, with knowledge alignment achieved through designed interaction and constraint mechanisms.

- We introduce the Dynamic Asynchronous Optimization (DAO) framework to address capability and optimization efficiency differences between modules. For capability differences, we design a dynamic knowledge alignment method to estimate module performance and adjust alignment strategies dynamically. For efficiency differences, we implement a proxy dataset as a knowledge transfer intermediary and design various asynchronous interaction strategies to ensure flexible and efficient joint optimization.

- We introduced a denoising module to enhance the robustness of the neural network model against data noise, improving the model's stability.

- Comprehensive experiments on four real-world memory behavior datasets demonstrate that SPsyINN outperforms state-of-the-art memory behavior modeling methods across all key metrics, and highlighting its potential for theoretical research and practical applications.

## 2 BACKGROUND

**Traditional Memory Theory Equations:** Memory modeling aims to explain and predict human memory (often referred to as forgetting) behavior using mathematical models. Early psychological studies predominantly relied on controlled experimental paradigms, analyzing data from such experiments to establish relationships between memory behavior features and memory performance, typically defined as the recall probability of specific memory materials.

The earliest research on human memory can be traced back to 1885 when Ebbinghaus proposed an approximate forgetting curve equation. He suggested that the interval since the initial memory event is a key factor affecting memory retention, which declines over time at a decreasing rate. This relationship was approximated using an exponential function. Subsequently, researchers explored other reasonable models for memory behavior. In 1974, Wickelgren (Wickelgren, 1974)proposed the generalized power law model ($R = \lambda(1 + \beta t)^{-\psi}$), where the recall probability ($R$) is modeled as a power-law function of initial memory strength ($\lambda$), time scale factor ($\beta$), forgetting rate ($\psi$), and time interval ($t$) since the last memory event. In 1995, Wozniak (Woźniak et al., 1995) introduced the dual-component model of long-term memory ($R = e^{-\frac{t}{S}}$), modeling recall probability ($R$) as an exponential function of memory strength ($S$) and time interval ($t$). In 2004, Anderson developed the ACT-R memory model ($R = \beta + \ln(\sum_{k=1}^{N} t_k^{-d_k})$) based on rational adaptation control theory for memory modeling. In 2009, Pashler (Pashler et al., 2009)proposed the MCM model ($R = \sum_{i=1}^{N} \gamma_i exp(-\frac{t}{\tau_i})x_i(0)$), suggesting that in repeated memory scenarios, memory performance is an aggregate of independent memory curves, similar to Wozniak's exponential model. In 2014, Lindsey introduced the DASH memory modeling method (Lindsey et al., 2014)($R = \sigma(a_s - d_c + \sum_{w=1}^{|W|}(\theta_{2w-1}ln(1 + c_w) + \theta_{2w}ln(1 + n_w)))$), which relates a learner's memory state ($R$) to their ability ($a_s$), material difficulty ($d_c$), attempt counts ($c_w$), and historical correct recall attempts ($n_w$). In 2016, the Half-Life Regression (HLR) model introduced the concept of memory half-life to describe the forgetting process of memory materials. Detailed explanations of the variables in these memory equations are provided in Appendix A.1.

Despite over a century of exploration, researchers have yet to identify a universally accepted memory equation. While theoretical memory equations are concise and interpretable, they have limited explanatory power for memory behavior and insufficient predictive accuracy for memory performance. Furthermore, many theoretical models include abstract psychological descriptors that are difficult to quantify. For instance, in Wozniak's dual-component model ($R = e^{-\frac{t}{S}}$), the descriptor memory strength ($S$) reflects the depth of impression left by a memory behavior on the learner. However, current research struggles to fully identify the factors influencing memory strength or to provide precise calculation methods, even though it clearly impacts memory performance. In practice, memory strength is often treated as a constant, which is evidently unrealistic. These issues pose significant challenges to building knowledge models based on psychological theories.

**Data-driven Parametric Model:** The widespread adoption of word memory software has opened new opportunities for memory research. Researchers have utilized data-driven paradigms and machine learning methods to develop parameterized memory behavior models, treating words as knowledge components in knowledge tracing (KT) (Bai et al., 2024). This approach integrates memory modeling with KT tasks, driving improvements in both model performance and theoretical insights. Advances in deep learning have further accelerated KT research. Piech et al. introduced the Deep Knowledge Tracing (DKT) model (Piech et al., 2015), the first to apply Recurrent Neural Networks (RNNs) (Lipton, 2015) to KT. DKT captures the temporal dynamics of student interactions with questions to predict responses to new ones, significantly outperforming traditional KT models and highlighting the potential of deep learning in modeling learning behaviors. Subsequent research adopted temporal models like Long Short-Term Memory (LSTM) (Ma et al., 2023; Sun et al., 2024) and Transformer (Liu et al., 2023), refining model structures (Sun et al., 2024) and incorporating factors such as difficulty levels (Han et al., 2013), review conditions (Shu et al., 2024), and material relevance (Chen et al., 2023) to enhance performance. While deep learning-based models excel in data fit and prediction accuracy, their "black-box" nature remains a challenge, limiting interpretability and educational applications. Moreover, building these models requires large-scale, high-quality behavioral data, which is still difficult to obtain.

**Physics Informed Neural Networks:** In recent years, Physics-Informed Neural Networks (PINNs) have emerged as one of the most successful knowledge-driven neural network models, achieving significant breakthroughs in fields like dynamics modeling (Hoffer et al., 2022), fluid mechanics (Wang et al., 2024a), and solving differential equations (Moseley et al., 2023). Unlike traditional purely data-driven neural networks, PINNs integrate domain-specific physical knowledge with deep learning, offering a novel approach to modeling. The core idea of PINNs is to embed physical laws (such as conservation laws and boundary conditions) directly into the neural network's loss function, ensuring that predictions and simulations always adhere to physical constraints. This approach not only enhances the physical interpretability of the model but also improves its generalization ability across various scenarios (Cuomo et al., 2022), demonstrating substantial potential for scientific computation and engineering applications. For example, the Navier-Stokes equations were applied to analyze the energy extraction efficiency of hydrokinetic turbines, while also improving the high-dimensional design of the turbine blades and ducts (Park et al., 2023).

However, in the domain of memory behavior modeling, the exploration of knowledge-informed neural network models remains scarce. Existing memory theory equations find it challenging to describe or predict real-world phenomena with the precision of physical equations. Their mathematical forms are often contentious, making it difficult to offer precise guidance to neural network models. Furthermore, psychological domain knowledge is challenging to express in computational models. Many abstract descriptors introduced in classical memory theory equations are difficult to translate into computable variables, posing significant challenges for knowledge representation.

## 3 METHODOLOGY

### 3.1 PROBLEM STATEMENT

We aim to develop and validate our approach using a large-scale word memory behavior dataset. Vocabulary Learning scenarios are widely used in memory behavior research (Meier et al., 2013), and the resulting memory models can guide Word Memorization Software in optimizing repetition strategies. These datasets are derived from real user interaction logs collected through Word Memorization applications. Users engage in word testing tasks provided by the software (as shown in Figure 1**a**), memorize target vocabulary, and retest the words after a certain period to reinforce memory. By analyzing learners' performance across different word tests over time, we can uncover the core patterns underlying their memory evolution and internal mechanisms. Our goal is to build computational models based on learners' historical interaction data from word tests, estimate their memory states for each word, and accurately predict their performance in upcoming tests for specific words (as shown in Figure 1**b**).

Formally, the set of all users in the dataset is denoted as $\mathcal{U} = \{u_1, u_2, \ldots, u_n\}$, and the set of all words as $\mathcal{W}$. The dataset encompasses all users' memory test behaviors, represented as $\mathcal{D} = \{\mathcal{D}_{u_1}, \mathcal{D}_{u_2}, \ldots, \mathcal{D}_{u_n}\}$, where $\mathcal{D}_u = \{[w_1, y_1, t_1], [w_2, y_2, t_2], \ldots, [w_m, y_m, t_m]\}$ denotes all behavior data for user $u$ in chronological order. Each behavior is described by a triplet $[w, y, t]$,

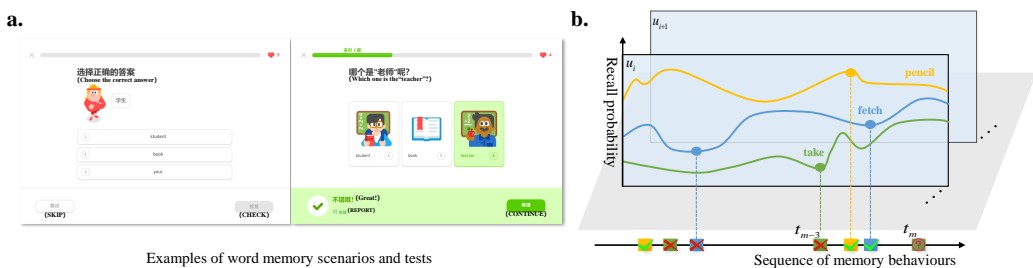

Figure 1: Memory Modeling Scenario: **a**. Learners engage in vocabulary review using various question types, such as multiple-choice, fill-in-the-blank, and listening exercises, as illustrated by the Word Memorization Software interface. Responses indicate their memorization state: correct answers signify successful retention, while incorrect ones imply incomplete memorization. **b**. The figure illustrates learners' performance across multiple review tests. Different colored curves represent memory retention trajectories for various words. The horizontal axis tracks testing performance over time, while the vertical axis denotes memory retention rates for specific words.

indicating user $u \in \mathcal{U}$ practiced word $w \in \mathcal{W}$ at time $t$ with a test outcome $y \in \{0, 1\}$, where $y = 1$ represents a correct response and $y = 0$ indicates failure, reflecting the user's memory state at that moment.

For a specific memory behavior $[w, y, t]$ of user $u$, we use $x_u^t$ to represent the historical memory behavior features of $u$ at time $t$, derived from all preceding behavior records. These features include six primary variables, whose definitions and computation methods are detailed in Appendix A.2. Correspondingly, $y_u^t$ denotes $u$'s performance on word $w$ in the memory test at time $t$. Our task is to build a memory model $f$ such that $y_u^t = f(x_u^t)$. All notations and definitions used in this paper are summarized and explained in Appendix A.3.

### 3.2 SPsyINN

We propose a Self-evolving Psychology-Informed Neural Network (SPsyINN), combining neural networks and genetic algorithms to design two independent modules: the Denoising Neural Network (DNN) and Genetic Symbolic Regression (GSR). The corresponding memory models are denoted as $f_{DNN}$ and $f_{GSR}$. Here, $f_{DNN}$ is a parameterized neural network trained via gradient-based optimization, while $f_{GSR}$ is a mathematical function optimized using genetic algorithms, with classical memory theory equations as the initial population.

To align the outputs of $f_{DNN}$ and $f_{GSR}$, we adopt techniques from knowledge distillation and PINN models, enabling knowledge integration while fitting training data. This chapter introduces the method in three parts: Denoising Neural Network, Genetic Symbolic Regression, and Dynamic Asynchronous Optimization. The first two sections detail the construction of $f_{DNN}$ and $f_{GSR}$, while the last explains their knowledge alignment and collaborative optimization. The overall framework is illustrated in Figure 2.

### 3.3 DENOISED NEURAL NETWORK

Neural networks, as universal approximators, have achieved significant success in behavior modeling, with temporal models like LSTM and Transformer widely applied. Our Denoised Neural Network (DNN) module adopts a classical learning behavior prediction architecture, combining a Temporal Neural Network (TNN) with a Multi-Layer Perceptron (MLP) classifier for modeling learners' internal states and classifying their performance. TNN can utilize flexible architectures such as LSTM (Hochreiter, 1997), Transformer (Vaswani, 2017), Mamba (Gu & Dao, 2023), or other specially designed model architectures.

For a learner $u$ with memory behavior data $D_u$, we concatenate all behavior features as $x_u^{t_{1:m}} = [x_u^{t_1}, x_u^{t_2}, \ldots, x_u^{t_m}]$, with the target memory performance sequence $y_u^{t_{1:m}} = [y_u^{t_1}, y_u^{t_2}, \ldots, y_u^{t_m}]$. The model's output predictions are $\hat{y}_u^{t_{1:m}} = MLP(TNN(x_u^{t_{1:m}}, \Theta_{TNN}), \Theta_{MLP})$, or, more generally

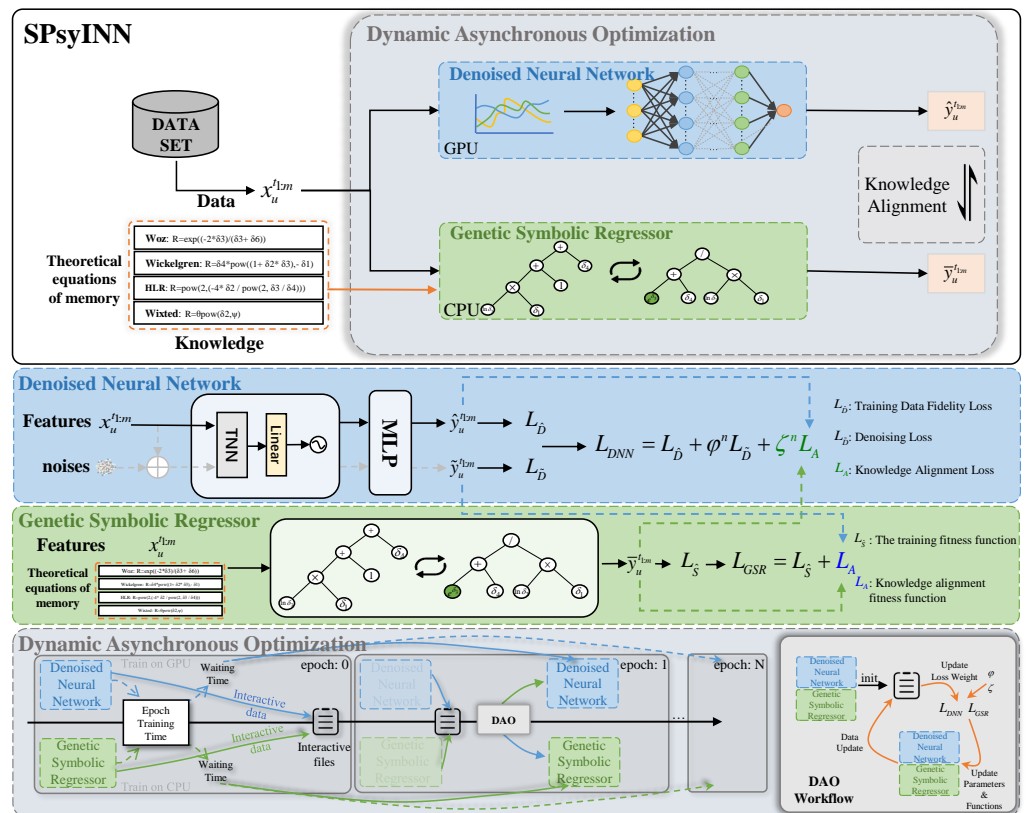

Figure 2: SPsyINN Model Framework Diagram. The blue subfigure describes the training process of the deep learning module; the green subfigure illustrates the training process of symbolic regression; the gray module represents the asynchronous training process.

$\hat{y}_u^{t_{1:m}} = f_{DNN}(x_u^{t_{1:m}}, \Theta_{DNN})$. The model optimizes parameters by minimizing the Mean Squared Error (MSE), $L_{\hat{D}} = \frac{1}{|\mathcal{D}|} \sum_{u \in \mathcal{U}} \sum_{i=1}^{m} (\hat{y}_u^{t_i} - y_u^{t_i})^2$.

To address noise in memory behavior data, we design a denoising module by injecting noise into the input features:

$$\tilde{x}_u^{t_{1:m}} = \sqrt{a_m} \cdot x_u^{t_{1:m}} + \gamma \cdot \varepsilon \cdot \sqrt{1 - a_m} \tag{1}$$

where $a_m = \prod_{t=1}^{m}(1 - \beta_t)$ is the cumulative noise schedule, $\gamma$ is a learnable noise weight, and $\varepsilon \sim N(0, I)$ represents Gaussian noise. This process is consistent with the perturbation kernel used in the Denoising Diffusion Probabilistic Models diffusion process (Song et al., 2020). A detailed proof can be found in Appendix B.1.

The model's noisy predictions are $\tilde{y}_u^{t_{1:m}} = f_{NN}(\tilde{x}_u^{t_{1:m}}, \Theta_{DNN})$, with the denoising objective to minimize: $L_{\tilde{D}} = \frac{1}{|D|} \sum_{u \in U} \sum_{i=1}^{m} (\tilde{y}_u^{t_i} - \hat{y}_u^{t_i})^2$. The DNN module's total training objective combines $L_{\hat{D}}$ and $L_{\tilde{D}}$, while overall optimization details are discussed in the Dynamic Asynchronous Optimization section.

### 3.4 GENETIC SYMBOLIC REGRESSOR

Genetic symbolic regression (GSR) is a classical symbolic regression algorithm that leverages evolutionary mechanisms of genetic algorithms to search for and optimize mathematical expressions, aiming to generate equations that meet specific requirements. The key steps include initializing a population, evaluating fitness, performing selection, mutation, and crossover operations, and updating the population. To incorporate insights from psychology, we use classical memory theory

equations as the initial population for GSR. The predictions from the GSR module are expressed as $\bar{y}_u^{t_{1:m}} = f_{GSR}(x_u^{t_{1:m}}, \Phi, \tau)$, where $\bar{y}_u^{t_{1:m}}$ represents the function values on raw data, $f_{GSR}(\cdot)$ is the optimized function derived from traditional memory equations, $\Phi \in \{+, -, \times, \div, \text{pow}, \exp, \ln\}$ denotes the operator set consistent with classical memory theories, and $\tau$ represents the current symbolic tree. The fitness function evaluates the GSR model's predictions and is defined as $L_{\hat{S}} = \frac{1}{|D|} \sum_{u \in U} \sum_{i=1}^m (\bar{y}_u^{t_i} - y_u^{t_i})^2$, ensuring the searched equations best fit the training data. Our GSR framework is flexible and supports various algorithms (e.g., TPSR (Shojaee et al., 2023), DGSR (Holt et al., 2023)) and libraries (e.g., Eureqa[1], PySR[2], and geppy[3]).

In summary, SPsyINN consists of two modules: a denoised neural network and a genetic symbolic regressor. Each module generates independent memory behavior predictions, namely $\hat{y}_u^{t_{1:m}}$ and $\bar{y}_u^{t_{1:m}}$, respectively. By default, we use $\hat{y}_u^{t_{1:m}}$(the output of DNN module) as the final output, as the denoised neural network typically achieves better prediction accuracy after training.

### 3.5 DYNAMIC ASYNCHRONOUS OPTIMIZATION

To align knowledge between the denoised neural network ($f_{DNN}$) and the genetic symbolic regressor ($f_{GSR}$) in SPsyINN, we propose the Dynamic Asynchronous Optimization (DAO) method for collaborative training. Knowledge alignment is achieved using the alignment loss $L_A$, defined as:

$$L_A = \frac{1}{|D|} \sum_{u \in U} \sum_{i=1}^m (\bar{y}_u^{t_i} - \hat{y}_u^{t_i})^2 \tag{2}$$

where, $\bar{y}_u^{t_i}$ and $\hat{y}_u^{t_i}$ represent the predictions of the symbolic regressor and the neural network, respectively. During training, a knowledge alignment objective is added on top of the data-fitting objective. This knowledge alignment loss function facilitates mutual learning, allowing weaker modules to benefit more from stronger ones. In the initialization phase, the randomly initialized $f_{DNN}$ primarily learns from $f_{GSR}$, which is grounded in theoretical equations. However, in later stages, if $f_{DNN}$ outperforms $f_{GSR}$, the alignment weight should be adjusted accordingly. Therefore, we propose a dynamic training objective adjustment method. The total loss for the neural network is:

$$L_{DNN} = L_{\hat{D}} + \varphi L_{\tilde{D}} + \zeta L_A \tag{3}$$

With dynamic weights $\varphi$ and $\zeta$ updated as $\varphi^{n+1} = \frac{L_{\hat{D}}^n + L_{\hat{S}}^n}{L_N^n + L_{\hat{S}}^n}$, $\zeta^{n+1} = \frac{L_{\hat{D}}^n + L_N^n}{L_N^n + L_{\hat{S}}^n}$. $L_N = \frac{1}{|\mathcal{D}|} \sum_{u \in \mathcal{U}} \sum_{i=1}^m (\tilde{y}_u^{t_i} - y_u^{t_i})^2$ represents the MSE loss of the neural network on noisy data. As $L_{\hat{S}}$ decreases, indicating improved fitting ability of $f_{GSR}$, the weight $\zeta$ increases, encouraging $f_{DNN}$ to learn more from $f_{GSR}$. Similarly, as $L_{\tilde{D}'}$ decreases, reflecting improved noise prediction by $f_{DNN}$, more emphasis is placed on $L_{\tilde{D}}$. For the symbolic regression module, the total fitness function is fixed as $L_{GSR} = L_{\hat{S}} + L_A$.

In implementation, $f_{DNN}$ and $f_{GSR}$ are trained as separate processes. For alignment loss computation, a proxy file serves as an intermediary. Predictions from each module for the corresponding batch are stored in this proxy dataset, which is updated after each epoch. Both modules read data from this proxy file to compute alignment loss $L_A$. The proxy dataset's batch sampling is independent of the training data batch sampling. Its batch size can differ from the training dataset, which, as supported by theoretical proofs (Appendix B.2), does not affect optimization performance. The overall workflow for DAO is illustrated in Figure 2.

In practical training, $f_{DNN}$ and $f_{GSR}$ often exhibit significant differences in training time per epoch, with DNN typically training much faster than GSR. This makes it crucial to enhance effective data exchange (knowledge alignment) between the modules. Additionally, the efficiency of knowledge alignment must be incorporated into the design, requiring a balance between alignment frequency and effectiveness. Our goal is to enable the two models to optimize synchronously, achieving better knowledge alignment results. Therefore, we have designed three types of asynchronous strategies to adjust the optimization pace between the two modules. **Wait Optimization (SPsyINN-W):** The

---

[1]http://nutonian.wikidot.com/

[2]https://astroautomata.com/PySR/

[3]https://github.com/ShuhuaGao/geppy

Table 1: The overall prediction performance of all baseline models and SPsyINN. The **best model performance** is in bold and the 2nd best is underlined(Excluding the variants of the proposed SPsyINN). * indicates t-test p-value $< 0.05$ compared to the 2nd best result. The experimental results for SPsyINN are based on predictions from the DNN module, with the reported values representing the averages of five independent experiments.

| Models | En2Es | | En2De | | Duolingo | | MaiMemo | |
|---|---|---|---|---|---|---|---|---|
| | MAE | MAPE | MAE | MAPE | MAE | MAPE | MAE | MAPE |
| Wickelgren | .1163 | 13.5908 | .1164 | 13.4275 | .1208 | 14.1423 | .2378 | 23.7749 |
| ACT-R | .1128 | 13.2630 | .1173 | 13.5106 | .1201 | 14.0784 | .2403 | 24.0310 |
| DASH | .1131 | 13.2853 | .1198 | 13.7429 | .1215 | 14.1991 | .2354 | 23.5378 |
| HLR | .1091 | 12.9216 | .1031 | 12.1911 | .1129 | 13.4130 | .2350 | 23.4967 |
| SBP-GP | .1176 | 13.5666 | .1218 | 14.1348 | .1108 | 13.3155 | .2660 | 26.6040 |
| PySR | .1112 | 12.9504 | .1125 | 12.9815 | .1180 | 13.8164 | .2238 | 22.8052 |
| DSR | .1225 | 14.1889 | .1499 | 16.5777 | .1293 | 14.9131 | .2394 | 24.2134 |
| TPSR | .6328 | 65.2455 | .7364 | 78.3057 | .5521 | 60.4980 | .3988 | 39.8840 |
| DKT-Forget | .1130 | 13.1517 | .1171 | 13.4490 | .1159 | 13.6450 | .2194 | 22.3509 |
| FIFKT | .1010 | 12.1402 | .1030 | 12.1639 | .1129 | 13.3785 | .2169 | 22.1095 |
| SimpleKT | .1070 | 12.6342 | .1115 | 12.9719 | .1079 | 12.8777 | .2266 | 23.1040 |
| QIKT | .1097 | 12.8703 | .1107 | 12.8539 | .1120 | 13.1363 | .2282 | 23.2759 |
| MIKT | .1092 | 12.7748 | .1105 | 12.8918 | .1128 | 13.3259 | .2313 | 23.4327 |
| SPsyINN-C | .0961 | 11.5564 | .0987 | 11.7130 | .0985 | 12.0220 | .2068 | 21.0379 |
| SPsyINN-I | .0923 | 11.2111 | .0959 | 11.4591 | .0965 | 11.9007 | .2071 | 21.0464 |
| SPsyINN-W | **.0922*** | **11.1970*** | **.0944*** | **11.3120*** | **.0901*** | **11.2543*** | **.2046*** | **20.7924*** |

faster module waits for the slower one to synchronize, ensuring alignment but reducing efficiency. **Continuous Optimization (SPsyINN-C):** Modules optimize independently, maximizing efficiency at the cost of alignment synchronization. **Interval Optimization (SPsyINN-I):** The neural network synchronizes every two epochs, balancing efficiency and alignment. Complete details of the algorithm and a clear diagram of the training process can be found in Appendix C.

## 4 EXPERIMENT

We conducted extensive experiments on four real-world datasets to verify the effectiveness of the proposed method. Thirteen benchmark models were introduced for comparison, including three categories: memory theory equations, symbolic regression algorithms, and deep learning methods. Detailed descriptions of the datasets, comparison methods, experimental criteria, evaluation metrics, and implementation details are provided in Appendix D.

### 4.1 COMPARISON EXPERIMENTS

To demonstrate the effectiveness of the proposed SPsyINN, we compared its prediction accuracy with 13 baseline methods on four datasets. The results are shown in Table 1. From Table 1, the following observations can be made: **1**.Compared to other baseline models, the proposed SPsyINN method significantly outperforms all benchmarks, demonstrating the validity of our approach and providing a novel framework and perspective for memory behavior modeling. **2**.Symbolic regression performs suboptimally, with methods such as SBP-GP (Pawlak et al., 2014) and TPSR (Shojaee et al., 2023) producing highly complex formulas that lose interpretability and often underperform compared to classical theoretical equations. We believe this is likely due to significant noise in the original data, which adversely affects the performance of symbolic regression models. **3**.Neural network models exhibit clear advantages over symbolic regression and memory theory equations because they can flexibly incorporate multiple complex attributes, automate relationship learning, and process these factors in parallel, enabling them to capture intricate interactions more accurately and provide more precise predictions. **4**.Experimental results with different waiting strategies (SPsyINN-C, SPsyINN-I, SPsyINN-W) show that per-round waiting (SPsyINN-W) achieves the best performance. Strategies with higher synchronization rates yield better performance but at the cost of lower training efficiency, requiring users to balance performance and efficiency when choosing an appropriate strategy.

## 4.2 ABLATION STUDY

To evaluate the impact of each module on the SPsyINN model, we conducted ablation experiments targeting one or two modules, including the Denoising module (DN, equation 1), Knowledge Alignment (KA, equation 2), and the Dynamic Weighting strategy (DW, equation 3) in the DAO method.

Table 2: Component Ablation experiments. A model without any selected components is referred to as TNN+Classifier, which excludes both the denoising and symbolic regression modules. Selecting the KA component indicates that the neural network and symbolic regression are jointly trained using the knowledge alignment method, and training is conducted using the waiting strategy. Selecting the DW component introduces dynamic weight optimization into the framework, where the loss weight of $L_{DNN}$ dynamically adjusts based on model performance. If DW is not selected, the loss weights remain preset constants. All values in the table represent the predictive performance of the neural network and the reported results are the averages of five independent experiments.

| Component | | | EN2Es | | En2De | | Duolingo | | MaiMemo | |
|---|---|---|---|---|---|---|---|---|---|---|
| DN | KA | DW | MAE | MAPE | MAE | MAPE | MAE | MAPE | MAE | MAPE |
| | | | .1130 | 13.1517 | .1171 | 13.4490 | .1159 | 13.6450 | .2194 | 22.3509 |
| | ✓ | | .1087 | 12.7599 | .1083 | 12.6222 | .1077 | 12.8895 | .2107 | 21.4313 |
| | ✓ | ✓ | .0974 | 11.6900 | .1019 | 12.0158 | .1051 | 12.6518 | .2086 | 21.2104 |
| ✓ | | | .0985 | 11.7919 | .1016 | 11.9960 | .0991 | 12.0817 | .2173 | 22.1103 |
| ✓ | ✓ | | .0968 | 11.6300 | .0968 | 11.5431 | .0953 | 11.7246 | .2065 | 20.9828 |
| ✓ | ✓ | ✓ | **.0922** | **11.1970** | **.0944** | **11.3120** | **.0901** | **11.2543** | **.2046** | **20.7924** |

Based on Table 2, the conclusions are as follows: **Symbolic regression improves performance**: Adding symbolic regression (second row) significantly enhances performance compared to the baseline DNN model (first row), demonstrating that symbolic knowledge can optimize neural network learning. **Synergy of knowledge alignment and dynamic optimization**: Models with both KA and DAO strategies (third and last rows) outperform those with only one (first and fourth rows), highlighting their combined effect in boosting accuracy and robustness. **Denoising module strengthens robustness**: Comparing the first and fourth rows shows that the denoising module significantly enhances the model's ability to handle noisy data. **Combining all components in the ablation study achieves the best results.**: The SPsyINN-W model (last row), incorporating all components (DN, KA, DW), achieves the best performance across all datasets.

To investigate the impact of initialization equations in symbolic regression, we conducted comparative experiments and expanded on more illustrative training details. The results indicate that initializing the GSR module with memory theory equations significantly enhances model performance. Furthermore, using a greater number of initialization equations further improves the outcomes, demonstrating the method's ability to effectively absorb and filter diverse prior knowledge while making the training process more stable. More detailed analyses are provided in Appendix E.1and E.2.

## 4.3 APPLICATION STUDIES

To explore our method's potential contributions to psychological theory, we conducted an in-depth analysis of the new theoretical equations discovered by the GSR module. By evaluating the predictive accuracy and form of these equations, we found that the equations produced by our method significantly outperform both classical theoretical equations and those generated by SR methods, demonstrating the advantage of our approach in identifying memory equations. Moreover, we observed that the memory state equations mined by SPsyINN not only remain consistent with traditional memory theory but also capture more complex behavioral relationships and interactions. These discovered memory states not only depend on time intervals but also include learners' historical learning performance, which may provide valuable empirical support for memory modeling. A detailed analysis process can be found in Appendix E.3. Additionally, we conducted numerical sensitivity analysis experiments, revealing that time interval-related variables in the Duolingo dataset exhibit higher sensitivity, indicating that memory states focus more on time interval information. In the MaiMemo dataset, memory states focus on both time intervals and learners' historical performance(see Appendix E.4 for detailed analysis).

Figure 3**a** illustrates the loss and corresponding evaluation metrics (Mean Absolute Percentage Error, MAPE) changes during the joint training of DNN and GSR. At the beginning of training, GSR performs well, so DNN gradually learns from GSR and optimizes parameters, dynamically adjusting loss and MAPE as training progresses. Eventually, DNN outperforms GSR. GSR continuously absorbs guidance from DNN, reducing MAE while effectively optimizing the equation form. The collaborative effect of DNN and GSR during training demonstrates the effectiveness and rationality of our proposed dynamic Asynchronous optimization method.

Figure 3**b** compares the memory equations mined by our model with traditional memory equations during long-term memory fits for a specific user on a particular word. As shown, our method accurately finds equations that better fit learners' memory effects, while traditional memory theory equations show poor fitting and over-predictions.

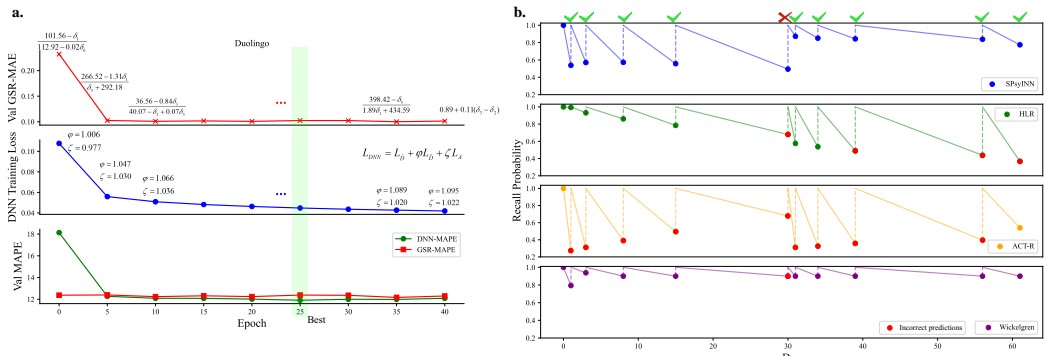

Figure 3: **a**. Dynamic training example of SPsyINN-W on the Duolingo dataset. **b**. Example of different memory equations predicting a learner's long-term memory effects (MaiMemo dataset).

## 5 CONCLUSION

We propose a novel psychologically interpretable dynamic asynchronous training model, SPsyINN, which effectively models memory behavior through knowledge injection and dynamic asynchronous optimization. Extensive experiments demonstrate that constraining neural networks with knowledge in memory scenarios is effective. Our framework enables efficient collaborative optimization of neural networks and symbolic regression, significantly improving the predictive performance of neural networks and the fitting accuracy of equations, thereby alleviating the issue of insufficient explanatory power of theoretical equations in memory scenarios. Methodologically, the dynamic alignment strategy enhances synergy, while in the asynchronous strategy, we observed a positive correlation between synchronization and model performance, though at the cost of training speed. In practical applications, SPsyINN reveals memory equations consistent with classical theories and identifies the dual influence of time intervals and learners' historical behaviors, offering valuable insights for memory modeling.

Future research will explore broader applications of SPsyINN, such as analyzing cognitive abilities like attention distribution and problem-solving, as well as applications in fields like cognitive science and finance. We aim to further enhance the model's generalizability, enabling it to integrate with other symbolic regression methods and offering a novel approach to scientific discovery.

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

## A    RELEVANT SUPPLEMENT AND DEFINITION

### A.1    DETAILS RELATED TO THE EQUATIONS OF MEMORY THEORY

For consistency, we represent memory retention as **Recall** ($R$) in the following memory equations.

- **Ebbinghaus** (Ebbinghaus et al., 1913): In 1885, the representation of the memory curve was proposed, indicating that the degree of forgetting is a functional relationship with the time elapsed without review. The ratio $b$ (representing how much of the first memory content is retained during the second memory attempt) of the time saved during relearning to the time initially spent learning is expressed as a function of the time interval $t$ between the first and second learning sessions, involving two parameters $c$ and $k$.

$$b = \frac{k}{(\log t)^c + k}$$

- **Wickelgren** (Wickelgren, 1974): Based on the traditional regression model of the generalized power-law memory model theory, the recall probability of memory material ($R$) is modeled as a power-law function of initial memory strength ($\lambda$), time scaling factor ($\beta$), forgetting rate controlling memory decay speed ($\psi$), and the time interval since the last memory ($t$). The equation is expressed as:

$$R = \lambda(1 + \beta t)^{-\psi}$$

- **Woz** (Woźniak et al., 1995): The dual-component model of long-term memory, modeling recall probability ($R$) as an exponential function of memory strength ($S$) and time interval ($t$).

$$R = e^{-\frac{t}{S}}$$

- **ACT-R** (Anderson et al., 2004): Based on the traditional regression model of adaptive cognitive characteristics and rationality, the learner's memory state ($R$) is expressed as a function of material difficulty ($\beta$) and the decay rate of memory strength ($d_k$) during the $k$-th review. The ACT-R memory equation is as follows:

$$R = \beta + ln(\sum_{k-1}^{N} t_k^{-d_k})$$

- **Wixted** (Wixted et al., 2007): Memory state $R$ is expressed as a power-law function of the review interval $t$, forgetting rate $\psi$, and a parameter $\theta$, and the equation is given as follows:

$$R = \theta t^{\psi}$$

- **MCM** (Pashler et al., 2009): The Multiscale Context Model (MCM) assumes that each practice session generates an exponential forgetting curve, and the forgetting process is approximated using a superposition of multiple exponential functions. The equation is as follows:

$$R = \sum_{i=1}^{N} \gamma_i exp(-\frac{t}{\tau_i})x_i(0)$$

Where $\gamma_i$ represents the scaling coefficients and $\tau_i$ represents the decay time constants, both of which are obtained through least-squares fitting to characterize the learner's memory strength.

- **DASH** (Lindsey et al., 2014): The learner's memory state ($R$) is expressed as a relationship with their student ability ($a_s$), material difficulty ($d_c$), number of attempts ($c_w$), and historical correct recall count ($n_w$). The DASH theoretical equation is as follows:

$$R = \sigma(a_s - d_c + \sum_{w=1}^{|W|} \theta_{2w-1} \ln(1 + c_w) + \theta_{2w} \ln(1 + n_w))$$

- **HLR** (Settles & Meeder, 2016): The half-life $h$ represents the time it takes for the learner's memory state to decay to $\frac{1}{2}$, and it is estimated using the word's features, the time interval between two reviews, the number of times the word has been encountered, and the number of times it has been correctly recalled. Here, $x$ is used to represent these features. The equation is expressed as follows:

$$R = 2^{-t/h}, h = 2^{\theta x}$$

## A.2 FEATURE DESCRIPTION

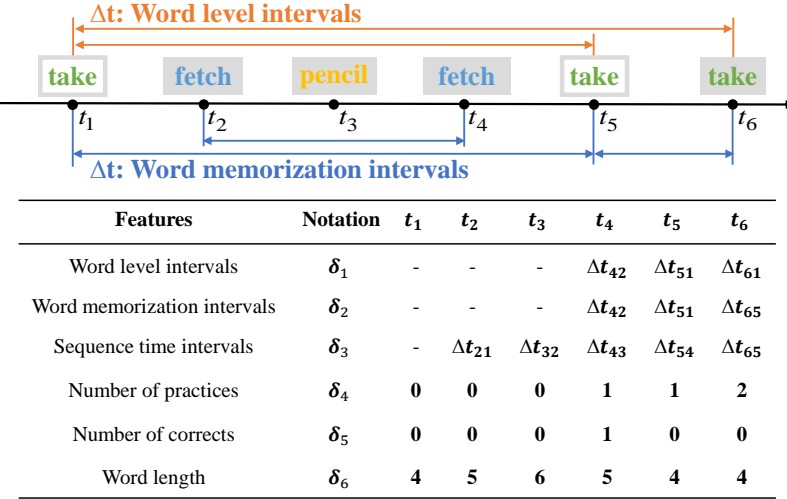

| Features | Notation | $t_1$ | $t_2$ | $t_3$ | $t_4$ | $t_5$ | $t_6$ |
|---|---|---|---|---|---|---|---|
| Word level intervals | $\delta_1$ | - | - | - | $\Delta t_{42}$ | $\Delta t_{51}$ | $\Delta t_{61}$ |
| Word memorization intervals | $\delta_2$ | - | - | - | $\Delta t_{42}$ | $\Delta t_{51}$ | $\Delta t_{65}$ |
| Sequence time intervals | $\delta_3$ | - | $\Delta t_{21}$ | $\Delta t_{32}$ | $\Delta t_{43}$ | $\Delta t_{54}$ | $\Delta t_{65}$ |
| Number of practices | $\delta_4$ | 0 | 0 | 0 | 1 | 1 | 2 |
| Number of corrects | $\delta_5$ | 0 | 0 | 0 | 1 | 0 | 0 |
| Word length | $\delta_6$ | 4 | 5 | 6 | 5 | 4 | 4 |

Figure 4: Description of Input Data Features.

We processed the raw data of the model into the following six features as inputs, with their visual representation shown in Figure 4.

- $\delta_1$: The interval between the learner's first memory of the word and the current timestamp.
- $\delta_2$: The interval between the learner's last memory of the word and the current timestamp.
- $\delta_3$: The interval since the learner's last memory activity, regardless of the consistency of memory material.
- $\delta_4$: The number of times the learner has reviewed the current word in prior memory activities.
- $\delta_5$: The number of times the learner has reviewed the current word in previous memory activities and successfully recalled it during testing.
- $\delta_6$: The length of the word, used as a simple descriptor of word difficulty.

During processing of the MaiMemo data, we did not obtain the $\delta_3$ data and only retained $\delta_1$, $\delta_2$, $\delta_4$, $\delta_5$, and $\delta_6$. In the Duolingo dataset, we use the complete features $\delta_{1:6}$. To account for the presence of certain review strategies in memory software, we standardized the above features using the training data set during model training.

## A.3 LIST OF SYMBOLS

We provide a detailed description of the symbols used in this paper in the table below.

Table 3: Nomenclature

| Symbol | Meaning |
| --- | --- |
| $\mathcal{D} = \{\mathcal{D}_{u_1}, \mathcal{D}_{u_2}, ..., \mathcal{D}_{u_n}\}$ | The dataset encompasses all users' memory test behaviors, represented as $\mathcal{D}$, where $\mathcal{D}_u = \{[w_1, y_1, t_1], [w_2, y_2, t_2], \ldots, [w_m, y_m, t_m]\}$ denotes all behavior data for user $u$ in chronological order. Each behavior is described by a triplet $[w, y, t]$, indicating user $u \in \mathcal{U}$ practiced word $w \in \mathcal{W}$ at time $t$ with a test outcome $y \in \{0, 1\}$, where $y = 1$ represents a correct response and $y = 0$ indicates failure, reflecting the user's memory state at that moment. |
| $x_u^t = [w, y, t]$ | The historical memory behavior features of $u$ at time $t$, derived from all preceding behavior records. |
| $y_u^t \in \{0, 1\}$ | The probability that user $u$ can recall the word at time $t$ during memorization, which serves as the label data for the model. |
| $f_{DNN}, f_{GSR}$ | The abbreviation for Denoised Neural Network and Genetic Symbolic Regression. |
| $\alpha_m = \prod_{t=1}^{m} 1 - \beta_t$ | Cumulative noise scheduling, where $\beta_t$ denotes the noise scheduling parameter, controls the intensity of noise added at each time step $t$. |
| $\gamma$ | It is a learnable noise weight used to adjust the degree of noise influence. |
| $\varepsilon \in \mathcal{N}(0, I)$ | Random noise generated according to a normal distribution. |
| $\tilde{x}_u^{t_{1:m}}$ | Learner interaction data after adding noise. |
| $\hat{y}_u^{t_{1:m}}$ | The predicted value obtained from the noiseless data through the neural network. |
| $\tilde{y}_u^{t_{1:m}}$ | The predicted value obtained from the noisy data through the neural network. |
| $\bar{y}_u^{t_{1:m}}$ | The predicted value obtained from the noiseless data through the symbolic regression. |
| $L_{\hat{D}} = \frac{1}{\|\mathcal{D}\|} \sum_{u \in \mathcal{U}} \sum_{i=1}^{m} (\hat{y}_u^{t_i} - y_u^{t_i})^2$ | The MSE loss between network's predictions on clean data and true labels |
| $L_{\tilde{D}} = \frac{1}{\|\mathcal{D}\|} \sum_{u \in \mathcal{U}} \sum_{i=1}^{m} (\hat{y}_u^{t_i} - \tilde{y}_u^{t_i})^2$ | The MSE loss between network's predictions on noisy and clean data. |
| $L_N = \frac{1}{\|\mathcal{D}\|} \sum_{u \in \mathcal{U}} \sum_{i=1}^{m} (\tilde{y}_u^{t_i} - y_u^{t_i})^2$ | The MSE loss between network predictions on noisy data and true labels. |
| $\Theta$ | The set of model parameters. |
| $\Phi = \{+, -, \times, \div, pow, \ln\}$ | The set of operators for symbolic regression. |
| $L_{\hat{S}} = \frac{1}{\|\mathcal{D}\|} \sum_{u \in U} \sum_{i=1}^{m} (\bar{y}_u^{t_i} - y_u^{t_i})^2$ | The fitness function is defined as the MSE between symbolic regression predictions and true labels |
| $L_A = \frac{1}{\|\mathcal{D}\|} \sum_{u \in U} \sum_{i=1}^{m} (\hat{y}_u^{t_i} - \bar{y}_u^{t_i})^2$ | The knowledge alignment loss is defined as the MSE between the neural network's predictions on clean data and those from symbolic regression. |
| $L_{DNN} = L_{\hat{D}} + \varphi L_{\tilde{D}} + \zeta L_A$ | The overall loss of the neural network. |
| $\varphi^{n+1} = \frac{L_{\hat{D}}^n + L_{\hat{S}}^n}{L_N^n + L_{\hat{S}}^n}, \quad \zeta^{n+1} = \frac{L_{\hat{D}}^n + L_N^n}{L_N^n + L_{\hat{S}}^n}$ | Dynamic Weights for Dynamic Asynchronous Asynchronous Optimization, where $n$ represents the iteration step. |
| $L_{GSR} = L_{\hat{S}} + L_A$ | The overall fitness function of symbolic regression. |

## B    RELEVANT PROOFS

### B.1    PRINCIPLES OF DIFFUSION PROCESSES

In the forward diffusion process of Denosing Diffusion Probabilistic Models (DDPM) (Song et al., 2020) , the perturbation kernel is defined as:

$$q(x_t|x_{t-1}) = \mathcal{N}\left(x_t; \sqrt{\alpha_t}x_{t-1}, (1-\alpha_t)I\right),$$

where $\alpha_t$ is the noise scheduling parameter at step $t$. Thus, the Markov property of the forward diffusion in DDPM can be expressed as:

$$x_t = \sqrt{\alpha_t}x_{t-1} + \sqrt{1-\alpha_t}\varepsilon_t, \quad \varepsilon_t \sim \mathcal{N}(0, I),$$

where the noise level at each step is controlled by $\sqrt{1-\alpha_t}$.

This result shows that the data at timestep $t$ in the forward diffusion process of DDPM is a linear combination of $x_0$ and noise $\varepsilon$, with weights determined by the cumulative noise factor $\bar{\alpha}_t$ and $1 - \bar{\alpha}_t$.

We extend DDPM by introducing a learnable noise weight, updating the perturbation kernel in DDPM as follows:

$$q(\tilde{x}_u^{t_{1:m}}|x_u^{t_{1:m}}) = \mathcal{N}\left(\tilde{x}_u^{t_{1:m}}; \sqrt{\alpha_t}x_u^{t_{1:m}}, \gamma^2(1-\alpha_t)I\right),$$

where $\gamma$ is a learnable noise weight that adjusts the noise component. With this perturbation kernel, the diffusion process can be described as:

$$\tilde{x}_u^{t_{1:m}} = \sqrt{a_m} \cdot x_u^{t_{1:m}} + \gamma \cdot \varepsilon \cdot \sqrt{1-a_m}$$

where, $\alpha_m = \prod_{t=1}^{m}(1-\beta_t)$, consistent with the cumulative noise factor in DDPM. When $\gamma = 1$, the diffusion process is fully equivalent to DDPM.

In our model setup, the learnable noise weight $\gamma$ provides the capability for dynamic noise level adjustment, enhancing adaptability and expressiveness across various tasks.

### B.2    THE PROOF OF OPTIMIZATION THROUGH SAMPLING PROXY DATA

Combining with the Monte Carlo approximation, we can express the loss $L_{DNN}$ after adding the previously defined alignment loss in the following process.

$$L_{DNN} = \frac{1}{|\mathcal{D}|}\sum_u\sum_{i=1}^{m}\left(y_u^{t_i} - \hat{y}_u^{t_i}\right)^2 + \varphi(\tilde{y}_u^{t_i} - \hat{y}_u^{t_i})^2 + \zeta(\hat{y}_u^{t_i} - \bar{y}_u^{t_i})^2$$
$$= \mathbb{E}_{(x_{u^*}^{t^*}, y_{u^*}^{t^*})\sim P(\mathcal{D})}((y_{u^*}^{t^*} - \hat{y}_{u^*}^{t^*})^2 + \varphi(\tilde{y}_{u^*}^{t^*} - \hat{y}_{u^*}^{t^*})^2 + \zeta(\hat{y}_{u^*}^{t^*} - \bar{y}_{u^*}^{t^*})^2)$$
$$+ \zeta(\hat{y}_{u^*}^{t^*} - \bar{y}_{u^*}^{t^*})^2)$$
$$\approx \mathbb{E}_{(x_{u^*}^{t^*}, y_{u^*}^{t^*})\sim P(\mathcal{B})}((y_{u^*}^{t^*} - \hat{y}_{u^*}^{t^*})^2 + \varphi(\tilde{y}_{u^*}^{t^*} - \hat{y}_{u^*}^{t^*})^2)$$
$$+ \mathbb{E}_{(x_{u^*}^{t^*}, y_{u^*}^{t^*})\sim P(\mathcal{PD}^{SR})}\zeta(\hat{y}_{u^*}^{t^*} - \bar{y}_{u^*}^{t^*})^2$$

Similarly, for $L_{GSR}$, we can also obtain the following process.

$$L_{GSR} = \frac{1}{|\mathcal{D}|}\sum_u\sum_{i=1}^{m}\left(y_u^{t_i} - \bar{y}_u^{t_i}\right)^2 + (\hat{y}_u^{t_i} - \bar{y}_u^{t_i})^2$$
$$= \mathbb{E}_{(x_{u^*}^{t^*}, y_{u^*}^{t^*})\sim P(\mathcal{D})}((y_{u^*}^{t^*} - \bar{y}_{u^*}^{t^*})^2 + (\hat{y}_{u^*}^{t^*} - \bar{y}_{u^*}^{t^*})^2)$$
$$\approx \mathbb{E}_{(x_{u^*}^{t^*}, y_{u^*}^{t^*})\sim P(\mathcal{B})}((y_{u^*}^{t^*} - \bar{y}_{u^*}^{t^*})^2$$
$$+ \mathbb{E}_{(x_{u^*}^{t^*}, y_{u^*}^{t^*})\sim P(\mathcal{PD}^{NN})}(\hat{y}_{u^*}^{t^*} - \bar{y}_{u^*}^{t^*})^2$$

where, $(x_{u^*}^{t^*}, y_{u^*}^{t^*})$ refers to data sampled from dataset $\mathcal{D}$. It is sampled according to the probability $P(\mathcal{D})$, where $P(\mathcal{D})$ represents a uniform distribution; $\mathcal{B}$ is generated directly from $\mathcal{D}$, representing the batch size of data when calculating the loss. $\mathcal{PD}$ represents a proxy dataset also generated from $\mathcal{D}$, and it combines the outputs of the neural network and symbolic regression to construct the corresponding proxy data $\mathcal{PD}^{NN} = [x_u^{'t_{1:m}}, y_u^{'t_{1:m}}, \hat{y}_u^{'t_{1:m}}]$ and $\mathcal{PD}^{SR} = [x_u^{'t_{1:m}}, y_u^{'t_{1:m}}, \bar{y}_u^{'t_{1:m}}]$. During model optimization, the DNN uses $\mathcal{PD}^{SR}$, while the GSR uses $\mathcal{PD}^{NN}$ .

## C ALGORITHM

**Continuous Optimization Strategy (SPsyINN-C):**This strategy seeks to maximize the optimization efficiency of each module by eliminating wait times. The modules operate independently, completing updates and reading the proxy dataset without waiting for each other, enabling fully asynchronous interaction. This strategy maximizes training efficiency but minimizes the frequency of effective interaction between the modules.

**Waiting Optimization Strategy (SPsyINN-W)**: This strategy aims to maximize the frequency of effective interaction between the models. The faster training module waits for the slower module to complete its training after finishing one epoch and then synchronously updates the proxy dataset before proceeding with the next round of training. This strategy maximizes the frequency of interaction but results in lower training efficiency with the longest wait times.

**Interval Optimization Strategy (SPsyINN-I):**In this strategy, the neural network interacts every two iterations, and the symbolic regression model interacts on every iteration.

---

Algorithm 1: SPsyINN-C

**Input:** The learner's word learning time series includes time, historical responses, word difficulty descriptions, target data $y$, number of epochs $N$, an initial set of traditional memory equations $f$.
**Output:** Trained model parameters and optimized memory equations.

1:    Initialize neural network parameters, initialize genetic symbolic regression with $f$
2:    **while** condition **do**
3:     **for** $i = 1 : N$ **do**
4:       Train neural network parameters and fit optimized memory equations
5:       Save the current optimal parameters and equations
6:       Sample $x_u'^{t_{1:m}}$ and save its predictions $\hat{y}_u'^{t_{1:m}}$ and $\bar{y}_u'^{t_{1:m}}$ as interaction data
7:       Read interaction data from the local file, and update loss weights, parameters,
        and the current optimal equation according to 3.5.
8:     **end for**
9:    **end while**
10:   **return** Parameters and optimized memory equations

---

Algorithm 2: SPsyINN-W

**Input:** The learner's word learning time series includes time, historical responses, word difficulty descriptions, target data $y$, number of epochs $N$, an initial set of traditional memory equations $f$.
**Output:** Trained model parameters and optimized memory equations.

1:    Initialize neural network parameters, initialize genetic symbolic regression with $f$
2:    **while** condition **do**
3:     **for** $i = 1 : N$ **do**
4:       Train neural network parameters and fit optimized memory equations
5:       Save the current optimal parameters and equations, and record the training time.
6:       Sample $x_u'^{t_{1:m}}$ and save its predictions $\hat{y}_u'^{t_{1:m}}$ and $\bar{y}_u'^{t_{1:m}}$ as interaction data
7:       Mutually read interaction data and update loss weights, parameters,
        and the current optimal equation according to 3.5.
8:       Based on the training time, the models wait for each other.
9:     **end for**
10:   **end while**
11:   **return** Parameters and optimized memory equations

---

| Algorithm 3: SPsyINN-I |
|---|
| **Input:** The learner's word learning time series includes time, historical responses, word difficulty descriptions, target data $y$, number of epochs $N$, an initial set of traditional memory equations $f$. |
| **Output:** Trained model parameters and optimized memory equations. |

  1:   Initialize neural network parameters, initialize genetic symbolic regression with $f$
  2:   **while** condition **do**
  3:    **for** $i = 1 : N$ **do**
  4:      Train neural network parameters and fit optimized memory equations
  5:      Save the current optimal parameters and equations
  6:      Sample $x_u^{'t_{1:m}}$ and save its predictions $\hat{y}_u^{'t_{1:m}}$ and $\bar{y}_u^{'t_{1:m}}$ as interaction data
  7:      GSR reads interaction data and updates the current optimal equation according to 3.5.
  8:      **if** $i \mod 2 == 0$ **then**
  9:        DNN reads interaction data from GSR and updates loss weights and parameters according to 3.5.
10:    **end for**
11:  **end while**
12:  **return** Parameters and optimized memory equations

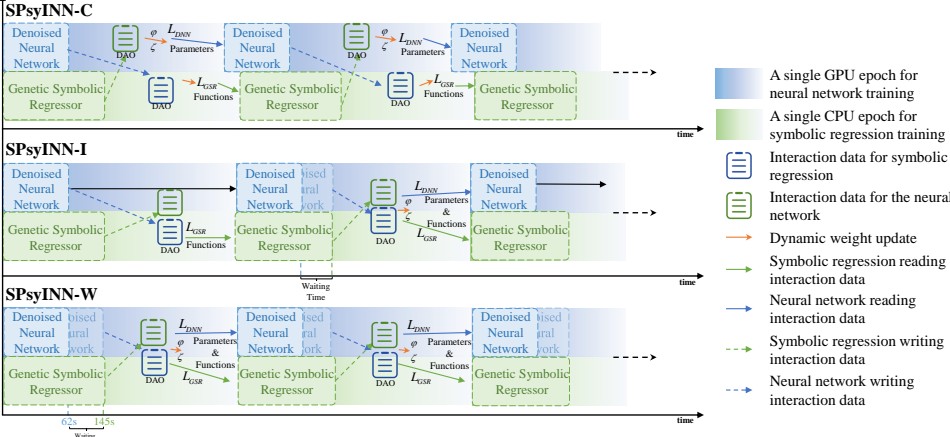

Figure 5: Schematic of Different Training Strategies.

# D    EXPERIMENT

## D.1    DATASETS

We evaluated the performance of SPsyINN on 4 publicly available commonly used datasets: Duolingo, Duolingo-En2De, Duolingo-En2Es and MaiMemo:

- **Duolingo**[4]**:** This dataset is a real-world dataset widely used in language learning applications. It contains language learning logs of 115,222 learners of English, French, German, Italian, Spanish, and Portuguese, recording 12.8 million student word courses and practice course logs.

- **Duolingo-En2De:** From the Duolingo dataset, we extracted user data of those learning German using the English UI interface and named it the Duolingo-En2De subset. This subset contains 117,037 log data points from 2,485 learners.

- **Duolingo-En2Es:** From the Duolingo dataset, we extracted user data of those learning Spanish using the English UI interface and named it the Duolingo-En2Es subset. This subset contains 271,854 log data points from 5,706 learners.

- **MaiMemo**[5]**:** This dataset comes from China's most popular English learning application "MaiMemo". It contains 200 million user learning records and 17,081 English words.

## D.2    BASELINES

To evaluate the effectiveness and robustness of our proposed SPsyINN model, we compare it with the following state-ofthe-art deep learning models and traditional theoretical models.

**Traditional Theoretical Equation Models**

- **Wickelgren** (Wickelgren, 1974): Based on the traditional regression model of the generalized power-law memory model theory, the recall probability of memory state ($R$) is modeled as a power-law function of initial memory strength ($\lambda$), time scaling factor ($\beta$), forgetting rate controlling memory decay speed ($\psi$), and the time interval since the last memory ($t$).

- **ACT-R** (Anderson et al., 2004): Based on the traditional regression model of adaptive cognitive characteristics and rationality, the learner's memory state ($R$) is expressed as a function of material difficulty ($\beta$) and the decay rate of memory strength ($d_k$) during the $k$-th review.

- **DASH** (Lindsey et al., 2014): The learner's memory state ($R$) is expressed as a relationship with their student ability ($a_s$), material difficulty ($d_c$), number of attempts ($c_w$), and historical correct recall count ($n_w$).

- **HLR** (Settles & Meeder, 2016): The half-life $h$ represents the time it takes for the learner's memory state to decay to $\frac{1}{2}$, and it is estimated using the word's features, the time interval between two reviews, the number of times the word has been encountered, and the number of times it has been correctly recalled.

**Symbolic Regression Model**

- **SBP-GP** (Pawlak et al., 2014): An improved genetic symbolic regression algorithm uses semantic backpropagation to heuristically invert the execution of evolving programs, optimizing the search process of operators.

- **PySR** (Cranmer, 2023): A symbolic regression tool based on genetic algorithms combines symbolic operations and simplification strategies to automatically generate interpretable mathematical formulas. It employs multi-objective optimization to balance model accuracy and complexity, features efficient parallelization, and is suitable for data modeling in scientific domains.

---

[4]https://www.duolingo.com/
[5]https://www.maimemo.com/

- **DSR** (Petersen et al., 2020): A reinforcement learning-based symbolic regression method trains a policy network to generate symbolic expressions.
- **TPSR** (Shojaee et al., 2023): A pre-trained Transformer-based symbolic regression planning strategy incorporates the Monte Carlo Tree Search algorithm into the Transformer decoding process, allowing non-differentiable feedback (such as fitting accuracy and complexity) to serve as external knowledge sources integrated into the equation generation process.

**Data-driven Parametric Model**

- **DKT-F** (Nagatani et al., 2019): An extension of Deep Knowledge Tracing (DKT) model (Piech et al., 2015) that incorporates a forgetting mechanism to predict user performance. The authors introduced three time-related features to improve the original DKT model: repetition interval, sequence interval, and the number of past attempts.
- **FIFAKT** (Ma et al., 2023): This model leverages an attention mechanism to dynamically integrate key information related to forgetting, question formats, and word semantic similarity, enabling more accurate predictions of user performance during the learning process..
- **SimpleKT** (Liu et al., 2023): By explicitly modeling question-specific variations and using a standard dot-product attention mechanism, the model captures individual differences in questions and time-related behavioral information, effectively addressing students' learning dynamics and changes in knowledge states.
- **QIKT** (Chen et al., 2023): Through question-sensitive cognitive representations and Item Response Theory (IRT) (Wilson et al., 2016), this model enhances the ability to model and interpret students' knowledge states, emphasizing the impact of question characteristics on their learning.
- **MIKT** (Sun et al., 2024): By simultaneously tracking students' domain knowledge states (coarse-grained) and concept knowledge states (fine-grained), and incorporating the Rasch (Rasch, 1993) representation method and IRT module, the model improves performance and interpretability, achieving multi-level modeling of students' knowledge states.

### D.3 EXPERIMENTAL SETUP

To train and validate the model, we used 80% of the student sequence data, reserving the remaining 20% for evaluation. All models were trained for 40 epochs using the Adam optimizer (Diederik, 2014) and repeated five times. An early stopping strategy was adopted: optimization was halted if the loss on the validation set did not improve within the last five epochs.

**Denoising Neural Network Architecture**

- **Base Structure:** LSTM with a three-layer linear MLP module.
- **LSTM:** Hidden layer size = 64, number of layers = 1.
- **MLP:** Linear(64 $\rightarrow$ 128)$\rightarrow$Tanh()$\rightarrow$ Linear(128 $\rightarrow$ 64) $\rightarrow$Tanh()$\rightarrow$ Linear(64 $\rightarrow$ 1) $\rightarrow$ Sigmoid()
- **Learning Rates:** MaiMemo dataset: 0.01; Duolingo dataset: 0.001
- **Batch size:** 256
- $\beta_t$**:**$\beta_t = linspace(0.001, 0.2, 100)$. Generate a uniformly distributed sequence of noise intensities ranging from 0.001 to 0.2 with a length of 100, for use in the noise module equation 1.

**Additional Notes for Ablation Study**  When KA is selected without DAO in SPsyINN, the loss function for DNN is expressed as:

$$L_{\text{DNN}} = L_{\hat{D}} + L_{\tilde{D}} + L_A,$$

with all weight coefficients set to 1, without dynamic adjustments.

**Details for Genetic Symbolic Regression**

- **Tool:** PySR
- **Settings:**
    - Population size = 40
    - Individual size = 50
    - Total iterations = 40
    - Cycles per iteration = 200
    - Maximum equation complexity = 15
    - Maximum nesting depth = 4

**Interaction Data Sampling:** 1024 samples.

All models were implemented in PyTorch and trained on a Linux server cluster equipped with NVIDIA GeForce GTX 2080Ti GPUs. Given the inconsistency in evaluation metrics between the Duolingo and MaiMemo datasets, we primarily used Mean Absolute Percentage Error (MAPE) as the main evaluation metric and Mean Absolute Error (MAE) as the secondary metric. The calculation methods for the evaluation metrics are as follows.

$$\text{MAPE} = 100\% * \frac{1}{|\mathcal{D}|} \sum_u \sum_{i=1}^m \left| \left( \hat{y}_u^{t_i} - y_u^{t_i} \right) / y_u^{t_i} \right|$$

$$\text{MAE} = \frac{1}{|\mathcal{D}|} \sum_u \sum_{i=1}^m \left| y_u^{t_i} - \hat{y}_u^{t_i} \right|$$

# E  ADDITIONAL EXPERIMENTS

## E.1  ANALYSIS OF THE IMPACT OF DIFFERENT INITIALIZATION EQUATIONS ON MODEL PERFORMANCE

Table 4: Performance of different initialization equations on four real-world datasets. The data comes from the predicted values of the DNN in the SPsyINN-C strategy. The reported results are the averages of five experiments.

| Models | En2Es | | En2De | | Duolingo | | MaiMemo | |
|---|---|---|---|---|---|---|---|---|
| | MAE | MAPE | MAE | MAPE | MAE | MAPE | MAE | MAPE |
| O-NN | .0985 | 11.7919 | .1016 | 11.9960 | .0991 | 12.0817 | .2173 | 22.1103 |
| SPsyINN(NO) | .0982 | 11.7583 | .1027 | 12.0887 | .0990 | 12.0734 | .2135 | 21.7136 |
| SPsyINN(ACT-R) | .1002 | 11.9475 | .1033 | 12.1483 | .0990 | 12.0728 | .2145 | 21.7949 |
| SPsyINN(HLR) | .0979 | 11.7300 | .1035 | 12.1670 | .0987 | 12.0437 | .2143 | 21.7766 |
| SPsyINN(Woz) | .0996 | 11.8860 | .1062 | 12.4250 | .0987 | 12.0451 | .2121 | 21.5675 |
| SPsyINN(Wick) | .0972 | 11.6710 | .1008 | 11.8940 | .0985 | 12.0270 | .2140 | 21.7651 |
| SPsyINN(Wixted) | .0982 | 11.7630 | .1018 | 12.0020 | **.0984** | **12.0160** | .2100 | 21.3499 |
| SPsyINN(ALL) | **.0961** | **11.5564** | **.0987** | **11.7130** | .0985 | 12.0220 | **.2068** | **21.0379** |

The definitions of various variants in the table are as follows:

- **O-NN(DNN)**: The proposed DNN model is trained independently without integrating GSR.

- **SPsyINN(NO)**: During GSR initialization, no equations are predefined. GSR directly searches for equations, which are then combined with the DNN using DAO for training.

- **SPsyINN(ACT-R)**: GSR is initialized with equations from ACT-R memory theory (Anderson et al., 2004), which are combined with the DNN and trained using DAO.

- **SPsyINN(HLR)**: GSR is initialized with equations from HLR memory theory (Settles & Meeder, 2016), which are combined with the DNN and trained using DAO.

- **SPsyINN(Woz)**: GSR is initialized with equations proposed by Wozniak's memory theory (Woźniak et al., 1995), which are combined with the DNN and trained using DAO.

- **SPsyINN(Wick)**: GSR is initialized with equations proposed by Wickelgren's memory theory (Wickelgren, 1974), which are combined with the DNN and trained using DAO.

- **SPsyINN(Wixted)**: GSR is initialized with equations approximating Wickelgren's memory theory proposed by Wixted (Wixted et al., 2007), which are combined with the DNN and trained using DAO.

- **SPsyINN(ALL)**: GSR is initialized with all the above memory theory equations, which are then combined with the DNN and trained using DAO.

The experimental results in Table 4 demonstrate that the SPsyINN (ALL) model, integrating five decay functions (ACT-R, Wozniak, HLR, Wixted, Wickelgren), outperforms others in most tasks. This highlights its superior ability to capture diverse memory decay patterns, crucial for handling complex, heterogeneous datasets. By combining multiple decay mechanisms, the model adapts to varied forgetting behaviors, aligning with cognitive science findings that memory decay involves multiple factors. This integration enables SPsyINN (ALL) to effectively model both rapid and slow forgetting processes, enhancing prediction performance across scenarios.

## E.2  ANALYSIS OF THE IMPACT OF DIFFERENT THEORETICAL MEMORY EQUATIONS ON NEURAL NETWORK TRAINING

Figure 6 provides key insights into loss performance: (1) SPsyINN-Wick and SPsyINN-Wixted show stable performance and fast optimization, consistent with Table 4. (2) DNN exhibits significant loss fluctuations, indicating that relying solely on neural network gradients is insufficient to escape local optima. (3) SPsyINN-NO, lacking equation initialization, is more volatile than SPsyINN-ALL, highlighting the stabilizing role of traditional memory equations in optimization. (4) On single-language datasets, SPsyINN-ALL underperforms in training error, while SPsyINN-Wick proves more stable. This suggests that memory equations reflect diverse memory states, a hypothesis supported by results on Duolingo and MaiMemo datasets.

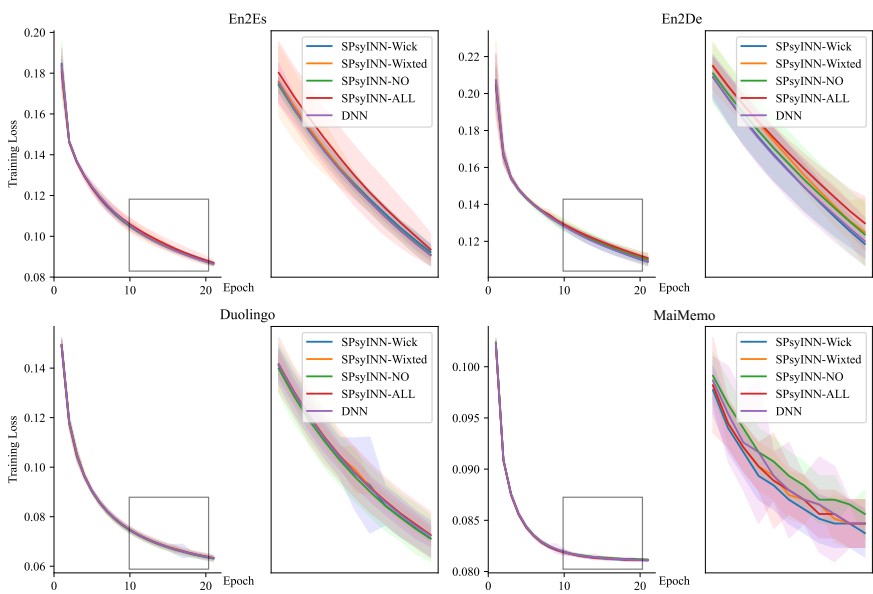

Figure 6: Loss performance of Asynchronous optimization training with different initialization equations on four datasets. The data comes from the predicted values of the DNN in the SPsyINN-C strategy.

### E.3    MEMORY EQUATION COMPARISON

**From the model performance perspective:**

- The proposed SPsyINN method outperforms traditional memory state equations in terms of performance and significantly outperforms the pure symbolic regression algorithm in terms of effectiveness.

- The symbolic regression algorithm based on genetic programming (such as PySR (Cramer, 2023) and different strategies of SPsyINN) can find relatively simple equations that align with memory theory. This is in contrast to symbolic regression algorithms based on reinforcement learning (DSR (Petersen et al., 2020)) or pre-training (TPSR (Shojaee et al., 2023)), which tend to generate more complex equation forms with lower theoretical interpretability.

- The SBP-GP (Pawlak et al., 2014) method performs well in general symbolic regression tasks (such as SRBench), but may not be suitable for domain-specific pattern mining tasks. This method typically generates equations with higher complexity during the search process, making interpretation more difficult, similar to the TPSR method.

**From the memory equation perspective:**

- **From the equation form:** The memory state equations mined by SPsyINN under different strategies mostly take the exponential form, which shares some similarities with the local form of HLR (Settles & Meeder, 2016). This similarity may indicate that memory states, to some extent, follow an exponential decay rule, as suggested by (Woźniak et al., 1995).

- **From the behavioral data features involved in the equation:** Similar to traditional memory theory equations, the memory equations mined by SPsyINN also highly focus on learners' time interval information $\delta_1, \delta_2, \delta_3$, which aligns with existing memory theory research (Randazzo, 2020). Particularly in the MaiMemo dataset, the equations mined by SPsyINN

Table 5: Memory Equation Comparison. The data in the table comes from the output of GSR. "-" indicates that the equation is too long to be displayed in full. All constants in the table are rounded to two decimal places. For precise values, please refer to our project.

| | Model | MAE | Function |
|---|---|---|---|
| **Duolingo** | **Wickelgren** | $.1208_{\pm.0005}$ | $0.89(1 + 0.0003\delta_2)^{-0.0003}$ |
| | **ACT-R** | $.1201_{\pm.0010}$ | $-0.56 \cdot \delta_6 + ln(\sum_{k-1}^{N} \delta_{2_k}^{0.05})$ |
| | **DASH** | $.1215_{\pm.0022}$ | $\sigma(1.89 - 1.14\delta_6 + \sum_{w=1}^{|W|} 0.1 \ln(1 + \delta_5) - 0.2 \ln(1 + \delta_4))$ |
| | **HLR** | $.1129_{\pm.0006}$ | $2^{\overline{3.53\delta_1 - 9.54\delta_2 - 0.2\delta_3 - 0.04\delta_4 - 0.18\delta_5 + 0.39\delta_6 - 0.06}}$ |
| | **SBP-GP** | $.1108_{\pm.0068}$ | - |
| | **PySR** | $.1180_{\pm.0034}$ | $0.90 - e^{\delta_1} + e^{\delta_2}$ |
| | **DSR** | $.1293_{\pm.0002}$ | $sin(exp(\delta_5 \cdot exp(\delta_2 - exp(\delta_4))))$ |
| | **TPSR** | $.5521_{\pm.0354}$ | - |
| | **SPsyINN-C-F** | $.1061_{\pm.0016}$ | $0.91 - (\delta_6 + \delta_2)(\delta_1 - \delta_2)$ |
| | **SPsyINN-I-F** | $\mathbf{.1020_{\pm.0008}}$ | $0.92^{(0.21 \cdot \delta_1 + \exp(\delta_6))}$ |
| | **SPsyINN-W-F** | $\underline{.1031_{\pm.0011}}$ | $0.93^{e^{\delta_6}}(\delta_5 + 0.02)^{(\delta_1 - \delta_2)}$ |
| **MaiMemo** | **Wickelgren** | $.2378_{\pm.0020}$ | $0.65(1 + 21.23\delta_2)^{-0.11}$ |
| | **ACT-R** | $.2403_{\pm.0006}$ | $0.84\delta_6 + ln(\sum_{k-1}^{N} \delta_{2_k}^{0.43})$ |
| | **DASH** | $.2354_{\pm.0025}$ | $\sigma(0.42 - 0.43\delta_6 + \sum_{w=1}^{|W|} 0.18 \ln(1 + \delta_5) + 4.08 \ln(1 + \delta_4))$ |
| | **HLR** | $.2350_{\pm.0033}$ | $2^{\overline{2.59\delta_1 - 4.38\delta_2 - 0.41\delta_4 + 0.37\delta_5 + 0.01\delta_6 - 0.05}}$ |
| | **SBP-GP** | $.2660_{\pm.0071}$ | - |
| | **PySR** | $.2238_{\pm.0027}$ | $0.21^{(\delta_1^2)^{(\delta_3 + 0.08)}}$ |
| | **DSR** | $.2395_{\pm.0002}$ | $cos(\delta_1 - \delta_5 + exp(\delta_1 \cdot \delta_2 \cdot \delta_4(-\delta_4 - \delta_5 - \delta_6) + \delta_6))$ |
| | **TPSR** | $.3988_{\pm.0626}$ | - |
| | **SPsyINN-C-F** | $\underline{.2215_{\pm.0039}}$ | $0.30^{(\delta_1 \cdot \delta_5 \cdot (\delta_4 + 0.14))}$ |
| | **SPsyINN-I-F** | $.2269_{\pm.0022}$ | $0.47^{(1.11 \cdot \delta_1)^{(\delta_4^{0.76})}}$ |
| | **SPsyINN-W-F** | $\mathbf{.2158_{\pm.0024}}$ | $0.49^{(\delta_1 + 0.01)^{(\delta_4^{0.63})}}$ |

place greater emphasis on the learners' historical performance features $\delta_4, \delta_5$, which is in strong agreement with the DASH theory (Lindsey et al., 2014).

- **From the interaction between behavioral data:** In the Duolingo dataset, the equation mined by SPsyINN-W introduces an additional term $(\delta_5 + 0.02)^{(\delta_1 - \delta_2)}$, which combines the historical performance $\delta_5$ with the time interval of the last memory $\delta_2$ and the time interval of the first memory $\delta_3$. Considering the full equation $0.93^{e^{\delta_6}}(\delta_5 + 0.02)^{(\delta_1 - \delta_2)}$, the factors influencing the current memory are jointly determined by material difficulty, review intervals, and historical performance. Additionally, the equation includes the term $(\delta_1 - \delta_2)$, representing the difference in time intervals between the first memory of a word and its last memory. This may reflect a chain memory effect in long-term memory processes, indicating that the associative impact between multiple memories may play a significant role in memory state modeling.

### E.4 EQUATION NUMERICAL SENSITIVITY ANALYSIS

Table 6: Sensitivity of Equation Coefficients and Variable Sensitivity. "-" indicates that the equation does not include the variable or the sensitivity of the variable is less than $1 \times 10^{-4}$.

| | Model | Function | Total-order indices | | | | | | | | |
|---|---|---|---|---|---|---|---|---|---|---|---|
| | | | $c_1$ | $c_2$ | $c_3$ | $\delta_1$ | $\delta_2$ | $\delta_3$ | $\delta_4$ | $\delta_5$ | $\delta_6$ |
| Duolingo | SPsyINN-C | $c_1 - (\delta_6 + \delta_2)(\delta_1 - \delta_2)$ | .1392 | - | - | .0013 | .3899 | .3983 | - | - | - |
| | SPsyINN-I | $c_1^{(c_2 \cdot \delta_1 + \exp(\delta_6))}$ | .9080 | .0031 | - | - | .0033 | - | - | - | .1014 |
| | SPsyINN-W | $c_2^{e^{\delta_6}}(\delta_5 + c_1)^{(\delta_1 - \delta_2)}$ | .6111 | .0873 | - | - | .2536 | .2531 | - | .2352 | .0877 |
| MaiMemo | SPsyINN-C | $c_1^{(\delta_1 \delta_5 (\delta_4 + c_2))}$ | .5555 | .0507 | - | .2278 | - | - | .1621 | .2315 | - |
| | SPsyINN-I | $c_1^{(c_2 \delta_1)^{(\delta_4^{c_3})}}$ | .9282 | .0325 | .0019 | .1461 | - | - | .0056 | - | - |
| | SPsyINN-W | $c_1^{(\delta_1 + c_2)^{(\delta_4^{c_3})}}$ | .8119 | .0510 | .0046 | .1947 | - | - | .0092 | - | - |

We performed Total-order indices sensitivity analysis on the equations generated by different strategies to assess the overall impact of input variables (or parameters) on the model output, including both the direct effects of variables and their interactions with other variables. Since the equations generated by the model are often in exponential form, the choice of base significantly affects the rate of change of the exponential function, making the base a key factor in determining sensitivity.

From the sensitivity analysis results, under the waiting strategy (SPsyINN-W), the sensitivity of the memory time interval was high in both the Duolingo and MaiMemo datasets, indicating its significant impact on memory prediction. However, there are differences in focus between the datasets:

- **Duolingo**: More focused on learners' time interval information, which is reflected in the higher sensitivity indices for time-related variables in the analysis results.
- **MaiMemo**: Shows more sensitivity to learners' historical performance, indicating that the model tends to adjust memory predictions based on past records.

Overall, these differences reflect the distinct characteristics of the datasets and further highlight the model's adaptability across different contexts.

