# OpenReview forum: "Combining Denoised Neural Network and Genetic Symbolic Regression for Memory Behavior Modeling via Dynamic Asynchronous Optimization"
_ICLR.cc/2025/Conference — Submitted to ICLR 2025_

### Official Review · Reviewer_CdzH · 2024-10-25

**Soundness:** 2
**Presentation:** 3
**Contribution:** 3
**Rating:** 5
**Confidence:** 5

**Summary:**

In this work, the Authors combine deep neural networks with genetic symbolic regression to model human memory in an efficient and interpretable manner. They proposed multiple ways to combine these two models, aiming at both compute efficiency and accuracy. The proposed model was tested on a panel of benchmarks where it showed an improved performance compared to a panel of baseline models.

**Strengths:**

The originality of the model lies in the novel combination of existing techniques (deep networks, denoising, and symbolic regression) and its application to the new domain (memory). I would like to especially highlight the new alignment algorithms, proposed here to account for the CPU – GPU interaction in the model, and domain priors that, first, kept the symbolic regression equations within the realm of memory model equations, and, second, accounted for the noise specific to memory-related data.

The significance of this model is in providing the equations (e.g. in Table 2) that are concise and at the same time have a high explanatory power for the memory-related data. Further analysis of such equations looks plausible and may be beneficial for the fields of psychology, cognitive science, and neuroscience.

The quality of this work is in the thorough empirical evaluation of the proposed model. While the model itself is rooted in literature and thus alone holds the potential of being useful for the task at hand, the evaluation on a series of datasets and the ablation study shows that the model indeed leads to the improvement of the performance and that all model components are necessary for such an improvement.

**Weaknesses:**

I would suggest working on the text a little bit more to enhance its clarity to make it more accessible to the broad ICLR community. While the work introduces a relatively straightforward idea – (i) merge an efficient deep learning model with an interpretable symbolic regression model and a denoising module; (ii) use auxiliary losses that would match the outputs of the three models; (iii) compute auxiliary losses on an intermittently-generated proxy dataset to smoothly synchronize CPU and GPU computations – the text itself is often unnecessarily complicated. For example, I’d either simplify Figure 2, remove the equations from it, or move it down the text to serve as a summary. I’d then simplify the equations and remove some of them because, while they introduce straightforward concepts – like, the mean square error loss – they end up being pretty lengthy. This is best exemplified by Equations 8 and 9 which say: “compute the loss on a batch” but somehow occupy nearly half a page. I would also draw attention to some of the results that are present in the paper but may be overlooked, e.g. the equations in Table 2. These results also could use further discussion. Besides, even though the code is provided, I couldn’t find the Methods section that would describe the model in sufficient detail to reproduce it (e.g. the parameters of the neural network and the training schedule).

Minor:

The background section mostly repeats the introduction. I’d suggest shortening one of these sections and either expanding the other with the details of the models or using the vacated space for an additional discussion of the results.

‘1 + 1 greater than 2’ effect -> synergy effect

KT-based framework: KT is not defined in the main text

MAE is not defined in the main text

Table 1: second best models: clarify that those do not include SPsyINN models

Table 2 is not referenced in the results.

Table 3: provide the statistical significance test data (ideally with the false discovery rate correction)

Overall, I found this work interesting and relevant, and I am happy to recommend it for acceptance at ICLR but I believe that the text needs to be further edited to enhance the accessibility of this work.

**Questions:**

The Authors state that the equations, that the model converges to, vary depending on the initial conditions and, it seems, on the model’s waiting strategies. Thus, which of the equations should neuroscientists / cognitive scientists use in their research as a result of this project? Are these equations similar or locally similar? Should they be distilled or approximated? How sensitive are these equations to the numerical coefficients? As one of the work’s stated goals is the interpretability of the results, it is important to know what results to use and to what degree to trust them. It would be great to hear the Authors’ thoughts on this topic. Separately, it would be highly interesting to see an analysis of the final equation once it’s established. How similar or dissimilar would it be to/from the existing models? What are the additional terms and what do we learn from them? Does it help us to ground the memory dynamics in neural circuits? An analysis like that has the potential to further increase the impact of this work.

_____________________________

Post-rebuttal: concerns mostly addressed (especially the ones regarding the clarity of the writing); raising my score to 8.

_____________________________

Post-discussion. We had a super lengthy and detailed discussion among the Reviewers where they encouraged me to check the reproducibility of the result. Sadly, they turned out to be right: (1) plugging the provided equations into the provided data reproduces the other Reviewer's numbers but not those in the paper; (2) MAE's denominator is not affected by zero labels; (3) zero labels cannot be excluded from a binary dataset.

As I mentioned before, I really like the paper but the other Reviewers are correct in pointing out that it's a serious issue. I hope that the Authors manage to revise their result towards consistency, stability, and reproducibility, hopefully also grounding them in cogsci-derived priors. Meanwhile, I sadly have to adjust my score to reflect the apparent reproducibility issue.

---

> ### Author Response · Authors · 2024-11-21
>
> Dear Reviewer CdzH,
>
> Thank you for your recognition of our work and for your valuable suggestions, particularly regarding the methodological innovation and empirical evaluation quality. We are delighted that you share an interest in the integration of neural networks with symbolic regression and acknowledge its potential in the field of memory modeling. As you noted, this research could inspire advancements not only in psychology, cognitive science, and neuroscience but also in other domains. Based on your suggestions, we have devised an improvement plan to enhance the clarity and readability of the manuscript, ensuring it is accessible to a broader academic audience. We plan to upload the revised version within 2-3 days and hope you will review it again and share your invaluable feedback.
>
> ### Improvement Plans for Identified Weaknesses:
>
> **1. Simplified formulas and illustrations**
> To enhance readability, we will simplify lengthy formulas (e.g., the definition of mean squared error), retaining only essential content. Descriptions of Equations (8) and (9) will be condensed to reduce space usage. Additionally, the equations in Figure 2 will be moved to the main text to better highlight the core concepts of the alignment algorithm.
>
> **2. Highlighting table content**
> The current discussion of results from Table 2, particularly those generated by symbolic regression, is insufficient. In the revised version, we will provide a more detailed analysis of these results, focusing on the potential implications of symbolic regression formulas for psychological and neuroscience research.
>
> **3. Additional details for model reproducibility**
> We will include more detailed information about the model in the revised manuscript, such as neural network hyperparameters, training schedules, and symbolic regression initialization settings. This will ensure other researchers can accurately reproduce our experiments.
>
> **4. Background and main text optimization**
> We recognize some overlap between the background and introduction sections. In the revised version, we will merge overlapping content, retain key background knowledge, and use the saved space to expand discussions on model details and experimental results.
>
> **5. Clarification of terms and abbreviations**
> We will clearly define all abbreviations upon their first appearance in the revised manuscript. Additionally, abbreviations not used repeatedly will be removed to avoid distracting readers.
>
> **6. Significance testing**
> For the experimental results in Table 3, we will add statistical significance testing data (e.g., standard errors or error ranges) to enhance the credibility of the results.
>
> **7. Enhanced referencing of Table 2**
> In the revised version, we will explicitly reference the equations and related discussions in Table 2, elaborating on their significance in memory modeling.

---

> > ### Author Response · Authors · 2024-11-21
> >
> > ### Detailed Responses to Questions:
> >
> > **1. Influence of initial conditions and waiting strategy**
> > As you mentioned, the final equations produced by the model are indeed influenced by the initial conditions and waiting strategy. In our experiments, different initial conditions may lead to symbolic regression producing equations in varied forms. However, these equations consistently capture core memory dynamics, such as forgetting speed and spacing effects. In other words, while they may differ in local characteristics, their overall trends and applicability remain stable.
> >
> > **2. Which equations should neuroscientists/cognitive scientists use as the results of this study?**
> > Our method generates a series of candidate equations. When selecting the final equation, we recommend prioritizing fit and simplicity while considering experimental design characteristics and relevant theories in the specific application context. This flexibility allows neuroscientists and cognitive scientists to select the most suitable formula for their needs.
> >
> > **3. Are the equations similar or locally similar? Should they be distilled or approximated? How sensitive are they to numerical coefficients?**
> > Our observations indicate that the equations generated by SPsyINN often follow exponential patterns and align with the spacing effects in memory theory. Additionally, we identified new insights in the memory equations, such as the influence of historical memory performance and material difficulty, which enrich the theoretical framework of memory modeling.
> >
> > We performed a sensitivity analysis on the numerical coefficients in the symbolic regression formulas. While noise can impact specific coefficient values, their influence on the overall trends of the formulas is minimal. The interpretability and predictive performance of these formulas remain stable across different datasets. In the revised version, we will include this analysis to better illustrate the robustness of symbolic regression.
> >
> > **4. Analyzing the final equations could be very interesting. How similar or different are they from existing models? What do the additional terms reveal? Can they help us model memory dynamics in neural circuits?**
> > In the revised version, we will provide a detailed analysis of the final equations, exploring their similarities and differences with existing memory models. Specifically, we will discuss additional terms identified by symbolic regression, which reveal potential new mechanisms in memory dynamics, such as the impacts of historical memory states and material difficulty.
> >
> > We believe these additional terms offer novel perspectives for studying memory dynamics in neural circuits. Through this analysis, we hope to further elevate the impact of this work.
> >
> > The revised manuscript will be uploaded to OpenReview within three days. We sincerely hope you will review it again. If you have further comments or suggestions, we would be happy to make additional modifications and improvements!

---

> > > ### Comment · Reviewer_CdzH · 2024-11-22
> > >
> > > Thank you for your detailed response. I'll stay tuned for the updated version.

---

> > > > ### Author Response · Authors · 2024-11-25
> > > >
> > > > Our revised manuscript has been uploaded to OpenReview for your review. If you have any additional feedback, we will make further refinements accordingly.

---

> > > > > ### Comment · Reviewer_CdzH · 2024-11-25
> > > > >
> > > > > Thank you for your effort.
> > > > >
> > > > > I've read the updated manuscript and the other reviews. My concerns have been addressed; most importantly; the clarity of the text has been much improved (clarity also has Reviewer qnLQ's concern). To the best of my understanding, the additional experiments provided by the Authors also address some of the Reviewer gLam's raised concerns regarding the benchmarks (although I would like to further discuss it with them).
> > > > >
> > > > > In the light of these improvements and the Authors' responsiveness, I raise my score.

---

> > > > > > ### Author Response · Authors · 2024-11-26
> > > > > >
> > > > > > Dear Reviewer CdzH,
> > > > > >
> > > > > > We sincerely thank you for your thoughtful comments and valuable questions, which have greatly contributed to the improvement of our work. Your recognition of our detailed responses, including the additional experiments and the revised manuscript, is deeply encouraging to us. We are especially grateful for your positive evaluation of our revised work and for raising the score to a clear 8.
> > > > > >
> > > > > > Once again, we deeply appreciate the time and effort you have invested in reviewing and helping us enhance our paper.
> > > > > >
> > > > > > Sincerely,
> > > > > > Authors of Paper 8572

---

> > > > > > > ### Author Response · Authors · 2024-12-01
> > > > > > >
> > > > > > > Dear Reviewer CdzH,
> > > > > > >
> > > > > > > Thank you for your meticulous review and valuable feedback during the rebuttal phase. Your insights have been immensely beneficial and have greatly helped us refine our work.
> > > > > > >
> > > > > > > Sincerely,
> > > > > > >
> > > > > > > Authors of Paper 8572

---

### Official Review · Reviewer_tRhP · 2024-10-30

**Soundness:** 3
**Presentation:** 3
**Contribution:** 3
**Rating:** 8
**Confidence:** 2

**Summary:**

The author propose a method for modelling memory behavior. This method combines symbolic regression and deep learning. A deep network and a symbolic regressor based on genetic algorithms are both jointly trained to predict memory performance from data. The neural network component also includes a loss based on noisy data to improve noise tolerance. Importantly, the two components are also trained to match each other through an alignment loss on their respective output, allowing both of them to train each other.

The resulting model seems to outperform existing approaches in predicting memory behaviors on various datasets.

===Edit after authors' response===

The authors have addressed my comments and largely clarified the paper. I believe the idea of jointly optimizing a neural network and an analytical expression is interesting and seems to show promise. Therefore, I increase my score, though still with low confidence.

**Strengths:**

The problem is interesting, as far as I can tell (I am not an expert in this area).

The method seems novel and the proposed interplay between deep learning and symbolic regression is interesting.

The model seems successful, judging from reported results.

**Weaknesses:**

I am mostly concerned about clarity, as the paper often uses confusing notation and undefined acronyms. While the overall method is reasonably clear, it is not easy to understand the details or reported comparisons in performance. See Questions below.

**Questions:**

The actual task is not fully described. Fig 1 mentions answers being "correct" or "incorrect", but about what? What was the actual question being asked for each word?

What is the final overall output of the model (i.e. the one used to generate results in the tables)? Is it the output computed from the generated equation, or the output of the neural network?

The notation is confusing and seems to vary. I'd recommend using always 1:m to denote multiple timesteps and m to denote one single time steps (in the last sentence of Problem statement, apparently just 'm' is used to denote a whole sequence?)

There are many undefined acronyms. E.g. in l. 269 What does KT stand for? Where does this "KT-based framework" come from?

L. 276, where do the Beta_t come from? The noise schedule equations look very much like the ones used in diffusion model, which should warrant some kind of citation!

In the results, particularly the ablations, the various alternative methods are not described. As a result it is not at all easy to understand what each alternative version represents. In particular: do you report results based on training a neural network alone (with or without denoising), and a symbolic regressor alone?

---

> ### Author Response · Authors · 2024-11-21
>
> Dear Reviewer tRhP,
>
> Thank you for your recognition of our self-evolving psychological embedding neural network model and for taking the time to review our submission. We apologize for the shortcomings in the paper’s presentation. We have carefully considered your suggestions and made comprehensive revisions to improve its clarity and structure. We plan to upload the revised version within 2-3 days and hope you can review it again to provide further valuable feedback. Your comments have been invaluable in helping us refine our work, and we are deeply grateful.
>
> ### Explanations and Responses to Weaknesses and Questions:
>
> **1. Incomplete description of the real-world task**
> We acknowledge that the original paper lacks clarity in describing the real-world task. In the revised version, we have clarified the problem setup with more detailed explanations and visual aids. Specifically, our study focuses on modeling learners’ memory states in a word memorization scenario. During the memory process, learners complete quiz questions through memory software. If a learner answers correctly, the word is marked as “mastered” (labeled as “correct”); if not, it is marked as “not mastered” (labeled as “incorrect”). Quiz formats include multiple-choice, fill-in-the-blank, listening, and matching questions. We have provided a more intuitive description of this task in the revised manuscript.
>
> **2. What is the model’s final output? Is it derived from the generated equations or the neural network?**
> The final output of the model is the prediction results from the neural network. All comparative and ablation experiments are evaluated based on the neural network’s output. The equations generated by symbolic regression are considered by-products of the model. In the equation comparison experiments, these equations are extracted and evaluated separately for their predictive performance. We have clarified the distinction between the two types of outputs in the revised manuscript to avoid confusion.
>
> **3. Confusing notation**
> We understand that the complexity of the notation may hinder readers' comprehension. To address this, we have thoroughly optimized the notation system in the revised manuscript. For example, we use $𝑚$ to represent a single time step and $1:𝑚$ to represent multiple time steps, and we have adjusted the mapping space of words to make the notation more intuitive and concise.
>
> **4. Undefined abbreviations**
> Thank you for pointing out the issue with undefined abbreviations. In the revised manuscript, we have resolved this by providing clear definitions for all abbreviations.
>
> **5. Source of $\beta_t$ and similarity to diffusion models**
> The noise scheduling parameter $\beta_t$ indeed shares similarities with noise generation equations in diffusion models. In the revised version, we have explicitly acknowledged this connection and added relevant citations. Additionally, we have included a detailed proof in the appendix to demonstrate the consistency between our noise generation method and the DDPM perturbation kernel.
>
> **6. Lack of clear descriptions of alternative methods in the ablation study**
> Our proposed method combines asynchronous training and dynamic optimization. Asynchronous training refers to the joint training of the neural network and symbolic regression model, while dynamic optimization adjusts the loss weights of the neural network dynamically. For instance, if asynchronous training is enabled without dynamic optimization, the model’s loss weights remain as fixed hyperparameters and do not change dynamically based on performance. It is important to note that dynamic optimization requires asynchronous training to be activated.
>
> In the ablation study, the configurations are as follows:
> - **Baseline model (DKT-F):** Does not include the denoising module or the symbolic regression module.
> - **Asy (asynchronous training):** Combines the neural network and symbolic regression with a waiting strategy during training, but the loss weights are fixed hyperparameters.
> - **DyOp (dynamic optimization):** Introduces dynamic loss weights that adjust based on model performance. If DyOp is not enabled, the loss weights remain fixed hyperparameters.
>
> In the revised manuscript, we have provided detailed explanations and refined the analysis of the experimental results.
>
> The revised version of our manuscript will be uploaded to OpenReview within three days. We sincerely hope you can review it then. If you have any additional suggestions, we would be delighted to incorporate them to further improve our work!

---

> > ### Author Response · Authors · 2024-11-25
> >
> > Our revised manuscript has been uploaded to OpenReview for your review. If you have any additional feedback, we will make further refinements accordingly.
> > As a point of clarification, in the ablation study section, we have renamed the components and provided detailed explanations to enhance clarity.

---

> > > ### Comment · Reviewer_tRhP · 2024-11-26
> > > **What is DKT?**
> > >
> > > In the baselines, the so-called "DKT" model is referenced multiple times, but it is never explained what DKT is (unless I missed it)! Please explain what "DKT" is.
> > >
> > > Also please fix the typography of citations (there should be a space before the first bracket).
> > >
> > > Considering the paper has been much improved and clarified, I will increase my score if these two corrections are applied.

---

> ### Author Response · Authors · 2024-11-27
>
> Dear Reviewer tRhP,
>
> Thank you for pointing out the insufficient explanation of Deep Knowledge Tracing (DKT) in our manuscript. In the revised version, we have added a detailed description of DKT in the Background section to enhance the clarity of this concept. Additionally, we have carefully reviewed all abbreviations in the manuscript to ensure their clear and consistent usage. Regarding the citation formatting issues you mentioned, we have corrected them in the latest revision. We sincerely appreciate your thorough review and valuable suggestions, which have been instrumental in improving the quality of our paper.
>
> Sincerely,
>
> Authors of Paper 8572

---

> > ### Author Response · Authors · 2024-12-01
> >
> > Dear Reviewer tRhP,
> >
> > Thank you for your meticulous review and valuable feedback during the rebuttal phase. Your insights have been immensely beneficial and have greatly helped us refine our work. We would be deeply grateful if you could consider providing a higher rating for our submission.
> >
> > Sincerely,
> >
> > Authors of Paper 8572

---

### Official Review · Reviewer_gLam · 2024-11-04

**Soundness:** 1
**Presentation:** 3
**Contribution:** 1
**Rating:** 3
**Confidence:** 5

**Summary:**

This paper proposes a new algorithm that jointly optimizes a neural network and an equation (that acts as an interpretable surrogate). The paper also explores update strategies of varying update rules. The approach proposed is then evaluated on real-world memory behavior datasets. The prediction performance of the neural network and discovered equation are reported separately.

**Strengths:**

1. The idea of jointly optimizing an equation with a neural network is novel among symbolic regression (SR) algorithms to the best of my knowledge.
1. Figure 2 gives a clear overview of the algorithm.

**Weaknesses:**

1. It is unclear whether the main aim of the paper is to discover memory equations or to propose a new SR-based algorithm. If the objective is to introduce both at the same time, then the paper is in an awkward position because it is not mentioned or made obvious in the paper why this specific task "to discover memory equations" require the proposed method (e.g., the paper should explain why the joint optimization with a neural network method is particularly effective for discovery memory equations and not applicable to other domains such as Physics). If the method proposed is indeed not specifically tailored to "discover memory equations", then evaluation on other datasets would provide a stronger case for this paper.

1. Missing comparisons to state-of-the-art SR algorithms, that do not use joint optimization with a neural network, should be included as comparisons in the paper (expand Table 2).

1. Existing SR benchmark datasets such as SRBench and SRSD should be used to evaluate the proposed algorithm's equation discovery ability to improve the quality of experiments.

1. Missing error bars for empirical results, unable to tell if the difference in performance is significant (apart from Table 1 which performs t-test).

1. Missing details on the selection of MLP architecture and tuning.

1. In line 308, PySR was selected among SR algorithms but this choice was not justified. Several recent state-of-the-art SR algorithms (e.g., DSR [1], TPSR [2]) should be considered as well. Otherwise, the paper should justify why these methods were not considered.

[1] Petersen, B. K., Larma, M. L., Mundhenk, T. N., Santiago, C. P., Kim, S. K., & Kim, J. T. (2020, October). Deep symbolic regression: Recovering mathematical expressions from data via risk-seeking policy gradients. In International Conference on Learning Representations.

[2] Shojaee, P., Meidani, K., Barati Farimani, A., & Reddy, C. (2023). Transformer-based planning for symbolic regression. Advances in Neural Information Processing Systems, 36, 45907-45919.

**Questions:**

1. How does SPsyINN perform on existing equation discovery benchmark datasets like SRBench and SRSD?

1. How do state-of-the-art SR algorithms perform in comparison on SPsyINN? The equations these state-of-the-art SR algorithms discover can be used to expand Table 2.

1. Where are the error bars for all the empirical results (i.e., standard deviation or inter-quartile range)?

1. Table 2 is given but not referenced to. Can the paper include a description and discussion of the results in Table 2?

Some of these questions may simply not be relevant because of the scope that the authors have set for the paper. If that is the case, I hope the justification for the limited scope can be addressed in the rebuttal.

**After Author-Reviewer Discussions:**

1.	The results are still not reproducible. I obtain the MAE values of 0.168, 0.164, 0.161 for PsyINN-C-F, PsyINN-I-F, PsyINN-W-F respectively, which differs largely from the values in Table 5. The standard deviation they have provided are in the range of 0.0016 to 0.0008, there is no reason for the values I obtained to be so different from what they have reported.

1.	The initial version, the first revision and the final revision has 3 separate set of equations. For example, for SPsyINN-I-F, MaiMemo, the discovered equation presented was different in all 3 versions. Given that the intention of these equations are meant for experts to analyze, I do not think the paper is in a ready-state given its frequent unstable updates.

1.	On closer inspection of the dataset, the true regression label is mostly 1. I computed the MAE of a naive regressor that always predicts the value 1, and obtained the MAE value of 0.1038 on the test set. This beats all but one of the methods in the duolingo dataset (that is if we can even trust the results. Based on my own re-computation of the equations in Table 5, none of their discovered equations beat this).

1.	MAE and MAPE are present, but because most of the values are 1, I also computed the R2 score which is the most common evaluation metric for equation discovery papers. The R2 score on duolingo were 0.00164, -0.00382 and 0.00774 for SPsyINN-C-F, SPsyINN-I-F, SPsyINN-W-F. These discovered equations will not be helpful to behavioural modelling.

**I recommend that the other reviewers do their own independent check on the reproducibility of the paper. This can be done quickly, purely in excel, just to check the equations, using 'test.csv' provided in the dataset (csv) file. After the many revisions, I do not trust using the evaluation code provided.**

I have not even had the time to check whether the “creation of these equations” are reproducible because of the constant revision of errors the paper made. I am considering downgrading my rating to "Strong Reject".

---

> ### Author Response · Authors · 2024-11-21
>
> Dear Reviewer gLam,
>
> We sincerely appreciate your detailed review of our submission and your recognition of the method integrating neural networks with equation optimization. Your valuable feedback has provided crucial guidance for improving our manuscript. We apologize for the shortcomings in the presentation of the paper. We have carefully considered your suggestions and revised the manuscript to enhance its clarity and presentation. We plan to upload the revised version within 2-3 days and hope you can review it again and provide further insights. Your comments have been instrumental in improving our work, and we are deeply grateful.
>
> ### Improvements and Explanations for Weaknesses:
>
> **1. Clarification of the paper’s objectives:**
> Our primary goal is to construct a knowledge-driven neural network model suitable for memory behavior modeling. To achieve this, we incorporate classical memory theory equations to constrain the neural network’s modeling process. However, existing memory theories are often debated and lack the explanatory precision seen in physical equations. Therefore, we initialize the neural network with classical memory theory equations and refine these equations using symbolic regression. This approach enables the neural network to absorb knowledge from classical memory theories while achieving collaborative optimization between the memory equations and the neural network model.
>
> In this process, the memory equations derived from symbolic regression serve as proxy models to explain the neural network and offer potential new perspectives for psychological theories. Our primary focus is not on proposing a novel symbolic regression method but on leveraging existing symbolic regression algorithms to address key challenges in memory behavior modeling.
>
> We chose genetic symbolic regression algorithms based on three considerations:
> 1. Genetic algorithms are a classic and well-researched family of symbolic regression methods, offering numerous resources and references.
> 2. They allow us to set initial populations, enabling classical memory theory equations to serve as initialization equations.
> 3. Genetic algorithms can strictly control the depth of symbolic trees, ensuring that the derived equations remain interpretable.
>
> Our framework is versatile and compatible with various genetic symbolic regression algorithms. PySR was selected as the codebase for our experiments, and we will clarify this in the revised manuscript. Additionally, we have added experiments to demonstrate the generality of our approach. Theoretically, our method also has the potential to integrate more advanced symbolic regression algorithms, which we plan to explore in future research to further enhance its applicability.
>
> **2. Lack of comparisons with state-of-the-art SR algorithms:**
> To address your suggestion, we have expanded the experimental section in the revised manuscript by including performance evaluations of advanced symbolic regression methods such as TPSR, DSR, and GP-SBP. These results have been incorporated into Table 2 to enrich the experimental comparisons.
>
> **3. Missing error bars in empirical results to assess significance:**
> Regarding the significance of empirical results, we have added error bars for all experimental results in the revised version and included statistical significance tests to enhance the credibility and completeness of the experimental findings.
>
> **4. Lack of details about the MLP architecture and adjustments:**
> We have provided detailed information about the architecture of SPsyINN in the revised manuscript, including the MLP architecture, hyperparameter settings, and training schedules, to ensure clarity and precision in describing the model.

---

> ### Author Response · Authors · 2024-11-21
>
> ### Responses to Specific Questions:
>
> **1. How does SPsyINN perform on existing equation discovery benchmark datasets?**
> Our primary research focus is on designing a model capable of effectively modeling learners’ memory states, rather than creating a general-purpose symbolic regression algorithm. Consequently, SPsyINN has not been evaluated on general equation discovery benchmarks (e.g., SRBench or SRSD). Nevertheless, we believe our proposed method holds potential for applications in other scenarios, and future research will explore its performance in broader domains.
>
> **2. How do advanced SR algorithms perform?**
> In response to your suggestion, we have included the performance of advanced symbolic regression methods such as TPSR, DSR, and SBP-GP in the memory modeling scenario. These results have been updated in Table 2 of the revised manuscript, providing a comprehensive comparison with SPsyINN to demonstrate the applicability of symbolic regression algorithms in memory modeling tasks.
>
> **3. Addition of error bars in experiments:**
> We have addressed this issue by including error bars (e.g., standard deviations or interquartile ranges) in the revised manuscript to highlight significant differences in the results.
>
> **4. Citations related to Table 2:**
> For the results in Table 2, we have added detailed discussions and included references to the theoretical equations involved in the revised manuscript, enabling readers to better understand the significance of the results.
>
> The revised version will be uploaded to OpenReview within three days. We earnestly request your review at that time and thank you once again for your thorough review and invaluable suggestions!

---

> > ### Comment · Reviewer_gLam · 2024-11-24
> >
> > The authors have clarified my concerns, and their changes should address all of my initial concerns. I am willing to increase my score, but am awaiting the upload of the revision to review the significance of their revised empirical results. I hope the authors can send a reply when the revision is uploaded so that I am notified, thanks.

---

> > > ### Author Response · Authors · 2024-11-25
> > >
> > > Our revised manuscript has been uploaded to OpenReview for your review. If you have any additional feedback, we will make further refinements accordingly.

---

> ### Comment · Reviewer_gLam · 2024-11-26
>
> The authors' revision has greatly improved clarity, so I will be increasing the presentation score to 3. The authors' have also identified key SOTA SR methods for comparison, which improves the quality of the experiments.
>
> However, I do have some new issues with the results. To verify the results of the paper, I am testing the obtained equations from Table 5 on the dataset given in the link. However, I am unable to reproduce the MAE scores of Table 5 (which is linked to Table 1). First, there seems to be an error for the results of SBP-GP (MaiMemo) in Table 5, where the average MAE is .3988 but .2660 in Table 1. Second, on the duolingo dataset, using the data extracted from the train_loader and test_loader (and also normalizing with the max and min code variables: "data = (data - min) / (max - min)", and changing nan values to 0: "data = torch.nan_to_num(data, 0)"), I get the following results:
>
> DUOLINGO dataset
> Wickelgren Model, "0.89*(1+0.0003*X[:,1])**-0.0003", MAE(using the provided "masked_mae" function provided by the authors): 0.1238
>
> PySR Model, "0.92 - X[:,0]*(X[:,4]+0.03)", MAE: 0.0987
>
> DSR Model, "np.cos(X[:,0]-X[:,3]+np.exp(X[:,0]*X[:,1]*X[:,2]*(-X[:,2]-X[:,3]-X[:,4])+X[:,4]))", MAE: 0.4667
>
> SPsyINN-C-F Model, "0.56*np.exp(-1.75*X[:,3]*X[:,0]*X[:,2])", MAE: 0.4150
>
> SPsyINN-W-F Model, "0.56*np.exp(X[:,0]*X[:,3]*(X[:,2]-X[:,0]))", MAE: 0.4149
>
> Could the authors', for the sake of easing the process for the reviewers to test reproducibility, extract a csv file with just the 6 features that are used for prediction (e.g., $\delta_1$, $\delta_2$, $\delta_3$, $\delta_4$, $\delta_5$, $\delta_6$), along with the output, R?
>
> Also, I have noticed in the code that these seem to be the normalized values based on the train set, I think for greater clarity, the authors should mention this in both Table 5 and Appendix A.2 for reproducibility. If $\delta_1$, $\delta_2$, $\delta_3$, $\delta_4$, $\delta_5$, $\delta_6$ refer to the unnormalized version instead, please correct me.

---

> > ### Comment · Reviewer_gLam · 2024-11-26
> >
> > To clarify, the csv should have all the relevant transformations so that the reviewers can directly plug the values into the equations present in Table 5.
> >
> > Currently, there is no documentation of the code, so I have done my best in reading through all the code and implementing the various data transformation, but I obtained the results as above. The provision of the csv is to the authors' benefit in helping the reviewers verify the reproducibility of the results. Also, if other reviewers can vouch for the reproducibility, I would appreciate if they could let me know if it is simply an error I made in processing the data/running the code.

---

> > > ### Author Response · Authors · 2024-11-26
> > >
> > > We sincerely apologize for the errors in Table 5, where some data were mixed up during formatting. Please note that the values in Table 1 are accurate.  Regarding the inconsistency in equation performance that you highlighted, this was an oversight on our part due to using the wrong equation. We have corrected this issue in the latest revised version for your review.  Additionally, we have updated the code repository to include a section on equation testing and uploaded the processed data directly. Our data were trained after applying a standardization procedure, and we have clarified this point in the revised manuscript—thank you for bringing this to our attention.
> > > As for your suggestion to provide a csv file for data extraction, we fully agree with its necessity. We will upload the file to the code repository within two days to further enhance the reproducibility of the tests.  We also plan to further optimize the code structure of the entire project to make it more organized and user-friendly.
> > >
> > > Thank you again for your valuable feedback.

---

> > > > ### Author Response · Authors · 2024-11-27
> > > >
> > > > Dear Reviewer gLam,
> > > >
> > > > We have uploaded the complete dataset in CSV format (dataset(csv)) to the code repository. These files include the corresponding $\delta_{1:6}$ and $Recall$ values, with all data preprocessed for direct use in symbolic regression to validate reproducibility. Additionally, we have provided a simple implementation of function performance testing (Test for functions). The data files involved in this process are also preprocessed to facilitate easier execution.
> > > >
> > > > We would like to clarify an important detail: in the MaiMemo dataset, $\delta_3$ is not available, and it was not used during model training. The features in the MaiMemo dataset consist only of $\delta_1, \delta_2, \delta_4, \delta_5, \delta_6$. However, for the Duolingo, En2De, and En2Es datasets, all features $\delta_{1:6}$ are included. This distinction has been explicitly stated in the submitted documentation.
> > > >
> > > > Thank you for your thorough review and for helping us improve the clarity of our work.
> > > >
> > > > Sincerely,
> > > > Authors of Paper 8572

---

> ### Author Response · Authors · 2024-11-30
>
> Dear Reviewer gLam,
>
> Thank you so much for taking the time to review our manuscript and providing such valuable feedback. Your comments have been immensely helpful in improving and refining our work.
>
> We have made every effort to address the issues and suggestions you raised, and we hope the revised manuscript meets your expectations. On this basis, we would like to kindly inquire whether, if you find our improvements satisfactory, you might consider raising the overall score for our paper.
>
> We greatly respect your professional judgment and would be happy to engage in further discussion if you have any additional questions or suggestions.
>
> Once again, thank you for your support and invaluable input throughout this process!
>
> Sincerely,
>
> Authors of Paper 8572

---

> > ### Author Response · Authors · 2024-12-01
> >
> > Dear Reviewer gLam,
> >
> > Thank you for your meticulous review and valuable feedback during the rebuttal phase. Your insights have been immensely beneficial and have greatly helped us refine our work. We would be deeply grateful if you could consider providing a higher rating for our submission.
> >
> > Sincerely,
> >
> > Authors of Paper 8572

---

> ### Author Response · Authors · 2024-12-04
>
> Dear Reviewer gLam,
>
> 1. **Regarding the inconsistency between equation performance and annotations**
> We believe there may have been some misunderstanding. We provided a simple implementation test for the performance of all equations (located in the code documentation under *Test for functions/Function_test.py*). We have carefully reviewed the composition of the CSV data and the performance of the equations, and we did not find any anomalies.
>
> We recommend reviewing the settings for equations in *Test for functions/Function_test.py*. We have carefully validated its reproducibility. Below are examples of the performance for specific equations:
>
> **MaiMemo Dataset**
> - **SPsyINN-C Function**: `0.30229160710443679**((delta1*delta5)**(delta4 + 0.14383100468725032))`
>   - Test: `MAE: tensor(0.2213)` | `MAPE: tensor(22.1332)`
> - **SPsyINN-W Function**: `0.49258729183071737**((delta1 + 0.008248828395980482)**(delta4**0.6295170754361378))`
>   - Test: `MAE: tensor(0.2157)` | `MAPE: tensor(21.5667)`
> - **SPsyINN-I Function**: `0.4733634187963817**((delta1**1.109281583119398)**(delta4**0.7567579728325409))`
>   - Test: `MAE: tensor(0.2251)` | `MAPE: tensor(22.5074)`
>
> **Duolingo Dataset**
> - **SPsyINN-C Function**: `-(delta6 + delta2)*(delta1 - delta2) + 0.9135621262263904`
>   - Test: `MAE: tensor(0.1052)` | `MAPE: tensor(12.6823)`
> - **SPsyINN-W Function**: `((delta5 + 0.0190478124994636)**(delta1-delta2))*0.9257066642765628**e**delta6`
>   - Test: `MAE: tensor(0.1041)` | `MAPE: tensor(12.5264)`
> - **SPsyINN-I Function**: `0.92171514151 ** (0.2101281541 * delta1+ exp(delta6))`
>   - Test: `MAE: tensor(0.1017)` | `MAPE: tensor(12.3638)`
>
> 2. **Regarding the inconsistency between the initial, revised, and final versions of equations**
> Our GSR module uses the PySR method, which provides a candidate set of equations during the symbolic regression process. We retained data files from each training session and selected the final equations from these files. When updating the equations, we rigorously confirmed their reproducibility. We have supplemented the code documentation with information on the equation sets for SPsyINN-C, SPsyINN-W, and SPsyINN-I for your review.
>
> 3. **Regarding the confusion about regression labels being 1**
> In the Duolingo dataset, this issue may arise from the dataset itself. Duolingo’s testing sessions allow learners only one incorrect response per word. Specifically, for a given word, testing stops only if the number of incorrect responses is less than one. This is explicitly reflected in the original dataset, and we observed this behavior. During model performance testing, we avoided equations and models that predict all values as 1. We have added examples of the original data to the code documentation to clarify this issue.
>
> 4. **Regarding the validity of the R² score**
> We have also considered this issue. While the identified equations perform well on MAE and MAPE metrics, their performance on the R² metric is poor. We believe this is likely due to the nature of the dataset. As you mentioned, most label values are 1, making it challenging for the equations to achieve a good R² score.
>
> We hope the above responses address your concerns. Once again, thank you for pointing out these issues.
>
> Sincerely,
> Authors of Paper 8572

---

> ### Author Response · Authors · 2024-12-04
>
> As a supplementary note, when calculating directly using the CSV data, the performance metrics must be computed in the format specified in the code documentation. This is because the label data contains zeros, which cannot be used as a denominator. We still recommend using the simple performance testing document we provided, as it offers a more convenient way for you to conduct the tests.
> We have set a mask in the calculation of metrics because there are 0 labels in the label data, and these would be treated as denominators in the metric calculations, which can lead to errors. This approach is consistent with the calculation process in FIFKT and is a standard practice in modeling. All standard calculation methods in our work are consistent.
> We have provided examples of how to calculate the corresponding metrics in the CSV.

---

### Official Review · Reviewer_qnLQ · 2024-11-04

**Soundness:** 3
**Presentation:** 1
**Contribution:** 3
**Rating:** 6
**Confidence:** 3

**Summary:**

This article presents a hybrid symbolic neural network learning approach to model user interactions in memory-based learning tasks, specifically language learning. The approach uses interpretable models based on memory theory and integrates their optimization with a neural network training process. A comparison with other methods for knowledge modeling demonstrate that this hybrid approach has performance benefits, beyond the creation of interpretable results.

**Strengths:**

The proposed methodology and the application domain are both interesting. The idea of training a neural network and a symbolic regression model simultaneously, and aligning them in their respective optimization processes, is worthwhile and potentially novel. It could be used in a number of applications where physical laws are known or, as is the case here, there are established equations that map the relationship that should be approximated by machine learning. The interpretability of the end result, combined with the performance of a neural network training, motivates this approach. It would be interesting to see the method applied to the discovery of known physical laws, like in the following work:

Cranmer, Miles, et al. "Discovering symbolic models from deep learning with inductive biases." Advances in neural information processing systems 33 (2020): 17429-17442.

The application of memory modeling is also interesting; I am not aware of the application of PINNs to such psychological modeling problems. Most of the knowledge-informed literature is on physics-informed, including in symbolic regression, but the methods can be applied to other domains where existing relationships have been expressed as equations, even if they are not physical laws. There is a clear application to economics here, and expanding the perspectives to include similar applications could be helpful.

**Weaknesses:**

The main weakness of this paper is in its presentation. This is a mix of methods and an application readers might not be familiar with, so everything, from deep learning to symbolic regression to memory theory, need to be made clear to a reader. Even a reader who is an expert in some of those things might not know the others.

First, the mathematical notation is highly verbose, with subscripts for almost all variables, even when certain information is redundant or clear. For example, all tasks map user data U to word sets W. The inclusion of this mapping for every variable is unnecessary and makes the loss equations, like Equations 8 and 9, very hard to parse. Some equations are maybe not even necessary, like the definition of MSE. Simplifying the notation and reducing redundancies in the math would greatly increase clarity.

More explanation of the baseline methods would also help. DKT-F and FIFKT aren't fully explained, nor is the way that symbolic regression is integrated into their methods. The one-sentence explanations in appendix B are not sufficiently nor sufficiently referenced in the main text. For example, the description "DKT-F: An improved version of DKT that incorporates students’ forgetting behaviors. (Piech et al., 2015)" assumes that the reader understands that DKT refers to "Deep Knowledge Tracing" (it was not defined), and that the reader is familiar with Deep Knowledge Tracing's standard mechanisms, which are not described. In the Background, a short explanation of at least DKT could easily replace the sentence "The superiority of deep learning techniques in knowledge tracing and cognitive modeling has been well-established (Abdelrahman et al., 2023)," which doesn't give much information to the reader and is rather subjective.

Greater clarity in the text on the background methods and the problem domain would really help. Acronyms are rarely defined before use, and some acronyms are defined never to be used again (eg, NODS). So, I'm left with a number of questions despite a thorough reading of the paper, which is a shame because it is a very interesting method and application.

**Questions:**

In the ablation, what does it mean to have asynchronous training but not "dynamic optimization"? And is the opposite of that (having dynamic optimization but not asynchronous training) the same as the Waiting Optimization Strategy SPsyINN-W?

How well does symbolic regression alone do, and is this equivalent to the first line of Table 3? If the neural network is trained without symbolic regression, how does it do, and is that equivalent to the fourth line of Table 3? Or is the fourth line equivalent to training a neural network without the noise addition? If either of those are the case, stating them in the text would be really useful.

Is the log in the function set a natural log? In table 8, the result of ACT-R is presented as a natural log, but the function set just says "log", which could be assumed to be log10. If it is log10, why not include ln if it is used in ACT-R's results?

---

> ### Author Response · Authors · 2024-11-21
>
> Dear Reviewer qnLQ,
>
> Thank you for recognizing the value of our work, particularly the innovative approach and potential of integrating neural networks with symbolic regression for memory modeling. Indeed, our research is greatly inspired by the PINN models from the field of physics, and we also believe our method has the potential for applications in other domains. We will mention this point in the revised manuscript. Additionally, we have included an analysis comparing the discovered memory equations with existing theories, highlighting our method’s potential in uncovering new psychological theories.
>
> We sincerely apologize for the shortcomings in the presentation of the manuscript, as noted by you and other reviewers. We have carefully considered your suggestions and have revised the manuscript to improve its clarity and presentation. We plan to upload the revised manuscript within 2-3 days and hope you can review it again and provide us with further valuable feedback. We truly appreciate your insightful comments, which have been immensely helpful in refining our work.
>
> ### Responses and Improvements to the Weaknesses:
>
> **1. Lengthy and redundant mathematical notations:**
> We acknowledge that the current version’s notations are overly complex, which may hinder the reader’s understanding. In the revised manuscript, we have simplified the notations and removed unnecessary redundancies. Furthermore, we have restructured the content to present mathematical symbols and formulas more clearly and concisely, reducing the cognitive load for readers.
>
> **2. Lack of clarity in baseline method descriptions:**
> Thank you for pointing out the inadequacies in the explanation of baseline methods. In the revised manuscript, we have enhanced the descriptions of baseline methods, including background details on the DKT-F and FIFKT models, and provided a thorough explanation of each variant in the ablation studies. We have also clarified the effects of combining different components, enabling readers to better understand the significance of various model configurations.
>
> **3. Enhancements to background methods and problem description:**
> Since the paper involves multiple approaches, including symbolic regression, deep learning, memory modeling, and PINNs, we have reorganized the manuscript’s logical structure and provided more detailed explanations of these background methods in the revised version. These changes will help readers gain a more comprehensive understanding of the research context and core contributions.
>
> ### Responses to Specific Questions:
>
> **1. Combination of asynchronous training and dynamic optimization:**
> Our proposed method integrates asynchronous training and dynamic optimization. Asynchronous training refers to the joint training of the neural network and the symbolic regression model in an asynchronous manner, while dynamic optimization involves dynamically adjusting the neural network’s loss weights. For instance, when asynchronous training is enabled but dynamic optimization is disabled, the loss weights remain fixed hyperparameters and do not change dynamically with model performance. It should be clarified that dynamic optimization requires asynchronous training to be enabled. We will further clarify these experimental settings in the revised manuscript.
>
> **2. Performance of symbolic regression when trained alone:**
> The primary objective of our study is to design a neural network model, with symbolic regression introduced to aid the neural network in better modeling learners’ memory states. In the original version, we did not provide the standalone performance results of the symbolic regression model. In the revised manuscript, we have added these results and analyses to provide a more comprehensive comparison.
>
> **3. Selection of operator sets in symbolic regression:**
> Thank you for raising this question. Due to an oversight, we did not explicitly specify the operator set used in symbolic regression. In this study, we adopted operators from traditional memory equations, and the $log$ function mentioned in the manuscript refers to the natural logarithm ($ln$). In the revised manuscript, we have corrected and standardized the symbolic descriptions to ensure consistent presentation and avoid confusion.
>
> Thank you once again for your valuable feedback. We will upload the revised manuscript to OpenReview within three days, and we look forward to your further review and suggestions.

---

> > ### Author Response · Authors · 2024-11-25
> >
> > Our revised manuscript has been uploaded to OpenReview for your review. If you have any additional feedback, we will make further refinements accordingly.

---

> > > ### Comment · Reviewer_qnLQ · 2024-11-27
> > >
> > > I thank the authors for the improvements made to the article. I do think that some presentation issues remain, but I have raised my score to a 6 to reflect the quality of the revised version. I’ll explain below some of the existing issues, which I didn’t take the time to raise individually in my initial review, but which continue my concern over clarity. My apologies for not having been more exhaustive in my first review, but I am heartened by the motivation of the authors to improve the article. An improved presentation is needed for a clear communication of the contribution of this article.
> > >
> > > Part of the issue in the presentation is prioritization. There is a fair bit of explanation given to the idea of asynchronous training, when the end result shows that synchronous training is preferable to asynchronous. It seems to me that the important part of 3.5 is knowledge alignment, as that impacts the results of the method more than the synchronicity of the training. DAO is presented as a part of the contribution, but in the end, if what works is to align the training epochs, the contribution of DAO is questionable compared to the overall contribution of the method. The ablation study of section 4.2, if I understand correctly, is entirely performed on the synchronous version of SPsyINN, for example. The contribution of SPsyINN without asynchronous training is interesting as it demonstrates the benefit of knowledge transfer - why insist on the asynchronous aspect, and why create a new acronym for it?
> > >
> > > A better focus would also help fix the priorities in the 10 page limit. Currently, the figures have been reduced to a very small size, making them almost unreadable on standard screens. Figure 2, for example, is more important than Figure 3, in my opinion, and if there is not the full place for both, then choosing one for the main text is far better than making both smaller.
> > >
> > > Furthermore, using established terms helps with presentation. I'm not familiar with any other works which refer consistently to symbolic regression (SR) as Genetic Symbolic Regression (GSR); SR is commonly used in the literature, whether it be deep symbolic regression or evolutionary. "Denoised" neural networks is also not a common term; the Denoising in DDPM refers to how the diffusion process denoises the image, for example. Here, prediction over memory data is denoised through a loss term - the neural network can be said to do a denoising operation, and the loss is appropriately called a denoising loss. The neural network, however, is not itself denoised, because the network doesn’t have noise. Finally, neural networks like LSTMs and Transformers can be applied to temporal data, but whether or not they are temporal themselves depends on if they are recurrent, hence why LSTMs are commonly referred to as an example of a Recurrent Neural Network, and not a Temporal Neural Network. For example, the authors state that “TNN can utilize flexible architectures such as LSTM (Hochreiter, 1997), Transformer (Vaswani, 2017), Mamba (Gu & Dao, 2023), or other specially designed model architectures,” but LSTMs and Transformers handle training and data representation differently due to the presence or absence of memory. How does the training change from a stateful RNN training to an ANN without state? If these seem like minor quibbles, I’ll point out that the title is currently “Combining Denoised Neural Network and Genetic Symbolic Regression for Memory Behavior Modeling via Dynamic Asynchronous Optimization”, and that I believe there are communication issues with “Denoised Neural Network,” “Genetic Symbolic Regression,” and “Dynamic Asynchronous Optimization.” These are central terms in the article, but they are not sufficiently grounded.
> > >
> > > Given that there has already been impressive improvements to the article over the review period, I would be open to further increasing my score. I realize that the paper revision deadline is approaching, but I’d appreciate responses from the authors up to the newly extended discussion period end date.

---

> > > > ### Author Response · Authors · 2024-11-28
> > > >
> > > > Dear Reviewer qnLQ,
> > > >
> > > > Thank you for your detailed feedback and for recognizing our work with an improved evaluation score. We have carefully reviewed and thoughtfully addressed all the issues you raised, making targeted revisions to our manuscript. Below, we provide specific responses to your major comments:
> > > >
> > > > **1. Priority Order of Topics**
> > > > In Section 3.5, we focused on *Knowledge Alignment (KA)* as it represents a significant innovation in our work. Regarding asynchronous training, our model components are deployed on GPUs (for DNN) and CPUs (for GSR), and their training processes are asynchronous. Here, "asynchronous" means the tasks performed by the models are not strictly sequential, with differing optimization speeds and steps.
> > > >
> > > > Specifically, the DNN predicts the learner's memory states, while the GSR focuses on identifying optimal equations to fit these states. These tasks are independent and executed concurrently, with integration achieved through Dynamic Asynchronous Optimization (DAO). To enhance collaboration between the models, we introduced *Alignment Loss* and an interaction strategy to ensure effective communication. While the interaction strategy includes a waiting mechanism that resembles synchronous optimization to some extent, it is distinct from strict sequential synchronization.
> > > >
> > > > In Section 4.2, regarding the ablation experiments, SPsyINN (including DN, KA, and DW) and the variant with only KA and DW both adopt a waiting strategy by default. Experimental results (Table 1) demonstrate that adequate interaction significantly enhances performance, supporting this choice in the ablation experiments.
> > > >
> > > > **2. Priority of Figures**
> > > > Thank you for pointing out the prioritization issue with our figures. We have accepted your suggestion by prioritizing Figure 2 and moving Figure 3 to Appendix C. Additionally, to enhance readability, we have replaced all figures in the manuscript with vector graphics and adjusted their dimensions to ensure clarity on standard screens.
> > > >
> > > > **3. Use of Common Terminology**
> > > >
> > > > *Symbolic Regression (SR):*
> > > > The symbolic regression algorithm in our work is based on Genetic Symbolic Regression, specifically PySR. PySR allows for flexible initialization of equations. Our work narrows the search space to enable evolutionary improvement starting from high-performing initial equations.
> > > >
> > > > *Denoised Neural Network (DNN):*
> > > > We apologize for the potential misunderstanding caused by the term "Denoised Neural Network." In our work, this term refers to a neural network with added noise, not actual denoising functionality. The noise addition aims to enhance robustness when handling noisy data, which is also the motivation for introducing the noise loss $L_{\tilde D}$. This term ensures consistency in the model's performance across noisy and noise-free data.
> > > >
> > > > We recognize that diffusion models like DDPM are commonly used in image denoising. Recent studies have extended diffusion effects to time-series modeling [1][2], highlighting its relevance in this context.
> > > >
> > > > *Temporal Neural Networks (TNN):*
> > > > TNN encompasses various time-series modeling methods, including RNNs (e.g., GRU, LSTM), CNNs, attention mechanisms, and Transformers. These methods are widely applicable to different modeling tasks, and the choice depends on the framework design.
> > > >
> > > > In our task, the goal is to extract memory states from learners' memory sequences and make predictions, and TNN is naturally well-suited for this purpose. Specifically, we designed a TNN that combines LSTM with MLP. During training, the MLP input includes both the LSTM's output and its hidden states, enabling efficient co-training from stateful RNNs to ANNs.
> > > >
> > > > We hope our responses address your concerns and provide a clearer understanding of our work. If you have any further questions or suggestions, we would be happy to incorporate them into further revisions. Thank you again for your valuable time and insightful comments!
> > > >
> > > > **References:**
> > > > [1] Yang L, Zhang Z, Song Y, et al. Diffusion models: A comprehensive survey of methods and applications[J]. ACM Computing Surveys, 2023, 56(4): 1-39.
> > > > [2] Kuo M, Sarker S, Qian L, et al. Enhancing Deep Knowledge Tracing via Diffusion Models for Personalized Adaptive Learning[J]. arXiv preprint arXiv:2405.05134, 2024.
> > > >
> > > > Sincerely,
> > > > Authors of Paper 8572

---

> > > > > ### Comment · Reviewer_qnLQ · 2024-11-28
> > > > >
> > > > > Dear authors, please provide your understanding of the word "asynchronous", as well as any published article which refers to symbolic regression as GSR or refers to all possible ANN architectures as TNN. I believe there are misunderstandings which persist.

---

> > > > > > ### Author Response · Authors · 2024-11-30
> > > > > >
> > > > > > Thank you for your meticulous review and invaluable suggestions on our work! Below are our detailed responses to the issues and concerns you raised:
> > > > > >
> > > > > > 1. On the understanding of the term "asynchronous''
> > > > > >
> > > > > > We sincerely appreciate your insightful comments regarding the usage and definition of "asynchronous.'' Upon reflection, we realize that our description of this concept may have lacked clarity, potentially causing confusion about the relationship between our waiting strategy and the asynchronous concept.
> > > > > >
> > > > > > Typically, "asynchronous'' refers to tasks or operations that can commence without waiting for preceding tasks to complete, characterized by non-blocking and concurrent behavior. In our study, the training of neural networks and symbolic regression models does indeed operate asynchronously and in parallel. However, under the waiting strategy, these two models temporarily pause after completing their respective training rounds to exchange the latest interaction data, aiming to enhance their collaborative effectiveness. After updating the data, both models resume the next iteration in parallel.
> > > > > >
> > > > > > From a synchronous perspective, this waiting strategy can be seen as the \(n\)-th epoch of the neural network waiting for the \((n-1)\)-th epoch of symbolic regression to finish before interaction occurs. Nevertheless, the \(n\)-th epoch of the neural network and the \(n\)-th epoch of symbolic regression execute in parallel. Hence, on a macro level, the training of the neural network does not completely halt or wait for the symbolic regression task to finish, distinguishing it significantly from traditional two-stage training methods.
> > > > > >
> > > > > > Our original design aimed to achieve fully asynchronous training (SPsyINN-C), where interaction data updates were implemented via local file reading mechanisms. However, in practice, the slower runtime of the symbolic regression algorithm introduced the following issues:
> > > > > >
> > > > > > For the neural network, it hindered timely access to more accurate aligned knowledge.
> > > > > > For symbolic regression, it failed to leverage better interaction data.
> > > > > >
> > > > > > This imbalance fell short of achieving the desired efficient collaboration. Consequently, we designed three interaction strategies (continuous optimization, interval optimization, and waiting optimization) to explore how a more effective knowledge alignment mechanism could enhance the synergy between the neural network and symbolic regression. We hope this explanation provides a clearer articulation of our understanding of "asynchronous'' and "synchronous'' and the rationale behind designing these strategies.
> > > > > >
> > > > > > 2. On uniformly naming symbolic regression as GSR
> > > > > >
> > > > > > Symbolic regression (SR) is a significant subfield of machine learning aimed at deriving symbolic mathematical expressions from data. Makke et al.[1] classify symbolic regression into five categories: Linear SR, Nonlinear SR, Expression-tree Search, Physics-inspired, and Mathematics-inspired. Genetic algorithm-based symbolic regression belongs to the Expression-tree Search category and is typically referred to as Genetic Programming Symbolic Regression (GPSR). In our paper, we abbreviate it as Genetic Symbolic Regression (GSR).
> > > > > >
> > > > > > The GSR mentioned in our paper specifically refers to the symbolic regression module in our model, which is implemented using PySR. As PySR is a genetic algorithm-based symbolic regression tool, we label it as GSR. We recognize that our terminology might have caused confusion in some parts of the paper. Upon careful review, we did not find instances where all symbolic regression methods were equated to GSR. For example, line 330 states:  "Our GSR framework is flexible and supports various algorithms (e.g., TPSR (Shojaee et al., 2023), DGSR (Holt et al., 2023)) and libraries (e.g., Eureqa, PySR, and geppy3).''
> > > > > > This was intended to emphasize the framework’s adaptability to different symbolic regression algorithms and tools rather than equating all symbolic regression to GSR.
> > > > > >
> > > > > > In the future, we aim to integrate advanced symbolic regression and neural network methods for multi-domain rule discovery tasks while maintaining greater rigor in terminology. Thank you for highlighting this concern!

---

> ### Author Response · Authors · 2024-11-30
>
> 3. On referring to ANN architecture as TNN
>
> In our paper, we use the term Temporal Neural Network (TNN) to refer collectively to neural network models designed for time-series data modeling (e.g., RNN, LSTM, GRU, Transformer). We acknowledge that this terminology, coined for convenience, may lack precision, and we deeply regret any confusion it may have caused.
>
> There are existing analogous naming conventions, such as Spatio-temporal Neural Network[2] and Temporal Convolutional Neural Network[3]. The use of TNN in our paper aimed to generalize a series of neural network models for time-series data modeling, including RNN, LSTM, Transformer, etc. Specifically, in our method, we employ LSTM to capture dynamic memory states and further process these states with an MLP.
>
> Once again, thank you for pointing out the potential confusion caused by this terminology!
>
> References
>
> [1] Makke N, Chawla S. Interpretable scientific discovery with symbolic regression: a review. Artificial Intelligence Review, 2024, 57(1): 2.
>
> [2] Ye J, Sun L, Du B, et al. Co-prediction of multiple transportation demands based on deep spatio-temporal neural network. In Proceedings of the 25th ACM SIGKDD international conference on knowledge discovery \& data mining. 2019: 305-313.
>
> [3] Pelletier C, Webb G I, Petitjean F. Temporal convolutional neural network for the classification of satellite image time series. Remote Sensing, 2019, 11(5): 523.
>
> We hope the above responses address your concerns clearly. Should you have further questions, we would be delighted to discuss them further! Thank you again for your support and suggestions regarding our work!

---

> > ### Author Response · Authors · 2024-12-01
> >
> > Dear Reviewer qnLQ,
> >
> > Thank you for your meticulous review and valuable feedback during the rebuttal phase. Your insights have been immensely beneficial and have greatly helped us refine our work. We would be deeply grateful if you could consider providing a higher rating for our submission.
> >
> > Sincerely,
> >
> > Authors of Paper 8572

---

### Meta-Review · Area_Chair_shvv · 2024-12-19

**Metareview:**

In my reading of this paper, the logic is as follows:
1. In the field of psychology, there are numerous proposed physical equations for human memory abilities.
2. It's not clear what the "right" equation is to fit empirical data, especially since certain variables in a given expression might also be outcomes of other expressions (e.g. $\exp(x / S)$ might have $S = ...$ another equation).
3. In order to find interpretable equations to fit empirical data, SPsyINN is proposed, which basically consists of a joint training procedure, in which:
    * An MLP must fit the empirical data
    * A symbolic regression expression must fit the empirical data
    * The MLP and the symbolic regression also must agree on most outputs.
4. Experiments were conducted on memory benchmarks, and shown to improve over other related baselines.

## Main Weakness
It's very unclear what the main contributions of the paper really are. I can think of two possible fronts, but each of them have their own drawbacks currently.
  * Proposal of a new general symbolic regression method.
    * It's not clear what the benefits of using the MLP really are. My interpretation is that it provides a form of smoothness regularization, to support regions of the input space where there were was no ground truth data. But there's not a strong justification for this design in the first place.
  * Application of deep learning methods to the specific field of memory equations.
    * It's not clear how significant these results are, and whether these applications are important for presentation at a conference such as ICLR, whose audience is primarily in machine learning. At the moment, the paper does not do a good job of providing the impact of said applications.

During the rebuttal, the authors emphasize that the regression technique and the dataset are inherently tied together and that they aren't proposing a general regression method, but in any case, there isn't a strong contribution from either direction listed above.

This, combined with the reviewer discussion, leads to a clear rejection score.

**Additional Comments On Reviewer Discussion:**

Being honest, this paper had the longest (and maybe _wildest_) discussion thread out of all papers in my batch. _Post-Rebuttal_, the reviewer scores post-rebuttal were across the table (3,5,6,8), with many reviewers in their own words, leaning towards rejection.

The main issues raised were:
  * What is the paper even contributing? As written also in my meta-review, the story of the paper is incredibly jumbled, especially combined with the authors' responses post-rebuttal. The paper doesn't do a good job of demonstrating (1) why their new symbolic regression method is good, or (2) demonstrating why memory equations is an impactful application.
  * Reproduction of results - Reviewer gLam has graciously spent their time trying to reproduce the results for Duolingo and Maimemo datasets, but could not. In fact, they have stated that the reproduced performance is even worse than a basic regressor which "only outputs 1", making the results untrustworthy.

I strongly suggest that the authors resolve these core issues before resubmitting to another conference.

---

### Decision · Program_Chairs · 2025-01-22

Reject